

# Sea ice local surface topography from single-pass satellite InSAR measurements: a feasibility study

Wolfgang Dierking[1,2], Oliver Lang[3], Thomas Busche[4]

[1]Alfred Wegener Institute Helmholtz Center for Polar and Marine Research, Bremerhaven, 27570, Germany
[2]Arctic University of Norway, Tromsø, 9019, Norway
[3]Airbus Defence and Space, Potsdam, 14467, Germany
[4]German Aerospace Center (DLR), Weßling, 82234, Germany

*Correspondence to*: Wolfgang Dierking (Wolfgang.Dierking@awi.de)

**Abstract.** Quantitative parameters characterizing the sea ice surface topography are needed in geophysical investigations such as studies on atmosphere-ice interactions or sea ice mechanics. Recently, the use of space-borne single-pass interferometric synthetic aperture radar (InSAR) for retrieving the ice surface topography has attracted notice among geophysicists. In this paper the potential of InSAR measurements is examined for several satellite configurations and radar frequencies, considering statistics of heights and widths of ice ridges as well as possible magnitudes of ice drift. It is shown that theoretically surface height variations can be retrieved with relative errors ≤ 0.5 m. In practice, however, the sea ice drift and open water leads may contribute significantly to the measured interferometric phase. Another essential factor is the dependence of the achievable interferometric baseline on the satellite orbit configurations. Possibilities to assess the influence of different factors on the measurement accuracy are demonstrated: signal-to-noise ratio, presence of a snow layer, and the penetration depth into the ice. Practical examples of sea surface height retrievals from bistatic SAR images collected during the TanDEM-X Science Phase are presented.

## 1 Introduction

Sea ice motion on scales of tens of meters to hundreds of kilometers changes as a function of time and space, dependent on variations of the forces exerted on the ice by wind, ocean currents, tides, and internal ice stress. Blocking of motion occurs along coastlines, around islands, and at other obstacles such as icebergs. The result is either a local opening of the ice or formation of ice ridges, rubble fields, and shear zones, leading to a steady change of the ice surface topography. In this article, the potential of interferometric synthetic aperture radar (InSAR) for measuring sea ice surface topography is addressed.

The shape and roughness of the ice surface determines the aerodynamic coupling between the ice and the atmospheric boundary layer (e. g. Garbrecht et al., 2002). Changes of surface height often indicate undulations of ice thickness, although ice depth changes do not necessarily mirror the surface undulations For example, the ridge keel is usually much broader than the ridge sail, and its depth exceeds the sail height by a factor between 4 and more than 10 (Strub-Klein and Sudom, 2012).



In some cases sail and keel do not occur concurrently (e. g. Tin and Jeffries, 2003). Nevertheless, ice thickness can be deduced from measurements of surface height variations using statistical approaches (Strub-Klein and Sudom, 2012, Petty et al., 2016). Another option is to measure the ice freeboard (the distance between the ice surface and the local water level) employing Cryosat-2 altimeter data, and from this to calculate the ice thickness, assuming hydrostatic equilibrium and

realistic ranges of ice density and snow mass load (Rickers et al., 2014). The determination of the ice freeboard is carried out at the margins of ice floes adjacent to open water leads, or to leads covered with thin ice.

Topographic measurements over sea ice have been carried out by means of helicopter-borne laser profilers (e. g. Dierking, 1995) or airborne laser scanners (Farrell et al., 2011). The relative height error of such sensors is on the order of 0.1 m, the footprint size between 0.1 m and 2 m, and the spatial sampling on ground ranges from 0.2 to 5 m. The largest of these values

approximately mark the upper limits that are necessary to resolve the surface height changes of, e. g., ice ridge cross sections with sufficient detail, considering the fact that the width of most ridges varies between less than a meter and 40 m, with only few exception reaching more than 70 m (Strub-Klein and Sudom, 2012). The laser altimeter on ICESat-2 (to be launched in late 2017) will have a 10 m footprint and an along-track sampling of 0.5 m (Farrell et al., 2011).

Until now, the majority of the published InSAR studies deal with data acquired over stationary ice (called "fast ice"). The

reason is that with the spaceborne systems employed in those studies (i. e. ERS, ALOS PALSAR, and Cosmo SkyMED), the necessary image pairs could only be acquired with temporal gaps of tens of hours to several days. In case of drifting ice, such time differences between subsequent orbits are much too large for achieving the magnitude of correlation between the two images that is necessary for a reliable interferometric height retrieval. Hence the investigations concentrated on indications of differential motion due to deformation processes in fast ice, links between ice properties and interferometric coherence,

and mapping of fast ice extent. (Dammert et al.,1998; Meyer et al., 2011; Berg et al., 2015).

The interferometric processing and height retrieval is based on the phase difference between two radar signals received from the same ground area element but from slightly different sensor positions. The geometric distance between the two sensor positions is called the baseline and consists of an along- and an across-track component ($B_{al}$, $B_{ac}$). The former is oriented parallel, the latter perpendicular to the satellite velocity vector. The along-track baseline causes a time lag between

the signal 1 and 2 received from a given surface element. This lag is denoted by temporal baseline and can vary from several days (repeat-pass InSAR) to a few microseconds (single-pass InSAR). An image showing the spatial variations of phase differences is called an interferogram. The phase difference can only assume values in the range from 0 to $2\pi$, which is usually represented by a matching color cycle in the interferogram. In worst cases, interferograms may reveal only noise-like patches indicating a total decorrelation between the received signals. Contiguous patterns of recurring color cycles called

fringes represent continuously increasing (or decreasing) phase differences between well-correlated signals. The interferometric phase difference $\Delta\phi$ is defined by (Madsen and Zebker, 1998):

$$\Delta\phi = \Delta\phi_{topo}(B_{ac}) + \Delta\phi_{mov}(B_{al}) + \Delta\phi_{noise} + 2\pi n \,, \tag{1}$$





This equation states that the measured phase difference may contain information about height variations of the images ground surface ($\Delta\phi_{topo}$) as well as about ground movements taking place between the reception of signal 1 and signal 2 ($\Delta\phi_{mov}$). The length of the across-track baseline determines the sensitivity to height variations, the length of the along-track baseline the sensitivity to ground displacements. The phase noise is caused by surface and volume scattering effects, by radar

system noise, and - in case of repeat-track InSAR - by atmospheric and ionospheric wave propagation delays. The last term takes into account that multiples of $2\pi$ may have to be added to the measured phase difference in further processing of the data (called "phase unwrapping"). Another important parameter that is determined from the image pair is the interferometric coherence, which represents the degree of correlation between both images.

Optimal conditions for retrieving sea ice topography and movement are given with single-pass InSAR when two antennas

are mounted on one satellite platform, or two satellites fly as a tandem in close formation. The opportunity to study the potential of single pass satellite InSAR for mapping of sea ice topography arose during the TanDEM-X Science Phase, which started in September 2014 and lasted for 17 months (Maurer et al., 2016). The TanDEM-X mission (TerraSAR-X add-on for Digital Elevation Measurements) has primarily been designed for topographic mapping of the Earth's land masses (Krieger et al., 2007). In standard operation mode the achievable relative accuracies are 2-4 m vertically (dependent on slope

of terrain and land cover type), and 3 m horizontally at a horizontal sampling of 12 m (Krieger et al., 2007). This mode is optimized for topographic mapping of the land surface, but is not sufficient for retrieving height variations of the sea ice surface. The Science Phase was initiated to demonstrate new products and applications such as digital elevation models with higher accuracies than in standard mode or measurements of ocean currents. It consisted of different sub-phases, among them a large cross-track baseline formation with mean along-track separation of zero that was initiated in March 2015. Data

takes were performed in a bistatic mode (see below). The comparatively large baselines in this phase translated to a very high sensitivity for object elevations on the order of decimeters. The data that are presented in this paper were acquired during the large cross-track baseline formation.

In section 2, relevant theoretical equations are introduced that are needed to assess whether a given SAR configuration is suitable for measuring the sea ice surface topography. The possible performance of satellite configurations that at present are

either operational or under discussion is investigated in section 3, succeeded by preliminary results of measurements from the TanDEM-X Science Phase (section 4). While ideal conditions are assumed in section 2, real-world factors that influence interferometric measurements over sea ice are investigated in section 5. Finally, the conclusions emphasize the major findings of this feasibility study.

## 2 Basic concepts

In this section, ideal conditions are assumed, i. e. the sea ice does not move, all parameters appearing in the relevant equations can be accurately determined, and the penetration depth of the radar signal into the ice is negligible. If the along-track baseline is zero, the interferometric phase is neither affected by ice drift. Biasing and disturbing factors and their effect on height retrievals are discussed in section 5. The potential to retrieve sea ice surface topography from single-pass InSAR





can be assessed by evaluating the height of ambiguity $h_a$ (which is the height difference related to one phase cycle, i. e. $\Delta\phi = 2\pi$) and the relative height error $\sigma_h$ (Madsen & Zebker, 1998):

$$h_a = \frac{\lambda H tan\theta}{pB_n}, \qquad (2)$$

$$\sigma_h = H tan\theta \frac{\lambda}{2\pi pB_n}\sigma_{\Delta\phi} = \frac{h_a}{2\pi}\sigma_{\Delta\phi} \qquad (3)$$

with $\lambda$ – wavelength, $H$ – orbit height, $\theta$ – incidence angle, $B_n$ – baseline perpendicular to the line-of-sight (the projection of $B_{ac}$ perpendicular to the slant range), and $\sigma_{\Delta\phi}$ – phase noise. The estimation of the latter is described later in the text. The factor $p$ equals 1 if one image is acquired in monostatic and the second in bistatic mode (e. g. in a tandem, where one satellite transmits and both receive in a synchronized operation mode), and $p$=2 if both images are acquired in monostatic mode (i. e. both satellites transmit and receive independently). A discussion of pros and cons of the monostatic and bistatic

mode can be found in Krieger et al. (2007). The incidence angle $\theta$ is the average of the respective incidence angles at scene centers valid for the acquisitions of image 1 and 2. A high sensitivity to topography is achieved when the ambiguity height is small. At first sight this implies that a large normal (perpendicular) baseline, a short wavelength, a low orbit, and a steep incidence angle are favorable conditions for the retrieval of topographic data. However, the normal baseline cannot be selected arbitrarily large, and for the incidence angle, additional dependencies have to be taken into account, as is shown

below.

The question concerning baselines achievable in space is discussed in section 5. Another limitation is caused by the nature of the surface and volume scattering mechanisms. The received radar signal is the coherent sum of contributions from different scattering objects that are arbitrarily distributed in the ground resolution cell. Because electromagnetic interactions between single scatterers are random, the backscattering intensity can vary significantly around the mean value (the

"speckle" effect). In case of a satellite tandem, the radar intensities measured over a given surface element from two different positions differ due to speckle. This difference is spatially randomly distributed. The critical baseline marks the total loss of correlation (i. e. the point of total decorrelation) between the two images from which the interferogram is generated. It is defined by (Madsen and Zebker, 1998):

$$B_{cn} = \frac{\lambda H}{p\Delta y cos^2\theta} \qquad (4)$$

where $B_{cn}$ is the critical perpendicular baseline, and $\Delta y$ is the single-look ground-range resolution. It is assumed that the surface slope equals zero and can hence be ignored. The critical baseline is defined as the baseline corresponding to a fringe rate of $2\pi$ per range resolution cell. Baseline decorrelation is less severe at longer radar wavelengths, finer spatial resolution, and larger incidence angles. Equation (4) is valid for the case that only surface scattering but no volume scattering takes place. It is emphasized here that two counteracting effects have to be considered: one the one hand a larger baseline increases

the height sensitivity but on the other hand decreases the coherence. Hence, there must be an optimal baseline, which is a trade-off between those counteracting effects. A way to calculate an optimal baseline is given below.





The relative accuracy of the retrieved heights of sea ice deformation structures depends on the phase noise, which is a function of the signal-to-noise ratio (SNR) and the baseline decorrelation. The phase noise can be expressed as a function of the interferometric coherence (Rosen et al., 2000):

$$\sigma_{\Delta\phi}^2 = \frac{1}{2N_L}\frac{1-\gamma^2}{\gamma^2} \tag{5}$$

Here, $N_L$ is the number of independent estimates (number of looks) used to derive the phase differences, and $\gamma$ is the interferometric coherence, which in the case of a single-pass system, along-track baseline of zero, and pure surface scattering is given by $\gamma=\gamma_G\gamma_N$ (Rosen et al., 2000; Bamler and Hartl, 1998). The first factor is the geometric baseline or surface correlation:

$$|\gamma_G| = 1 - \frac{B_n}{B_{cn}} \quad B_n \leq B_{cn} \tag{6}$$

In the derivation of (6) it was assumed that the system transfer function is rectangular, and that no spectral shift filter is applied. The correlation as a function of system noise is:

$$\gamma_N = \left(1 + \frac{1}{SNR}\right)^{-1} \tag{7}$$

This equation is valid if the noise in image 1 is independent of the noise in image 2, and both noise levels are of same magnitude.

Considering the balance between a large sensitivity to surface elevation changes (i. e. the most favorable value of $h_a$) and the baseline decorrelation, the optimal baseline that minimizes the height error can be obtained from equation 8, which is the result of combining equations 2, 3, 4 and 5:

$$\sigma_h = \frac{\Delta y \, sin\theta \, cos\theta}{2\sqrt{2}\pi} \frac{\sqrt{\gamma_N^{-2}-(1-x)^2}}{x(1-x)} \tag{8}$$

where $x = B_n/B_{cn}$, and it was assumed that $N_L = 1$. By evaluating the derivative of equation 8 for $\gamma_N =1$, it is found that the
optimal normal baseline is $B_n = aB_{cn}$ with $a$=0.382. For $\gamma_N < 1$, the factor $a$ increases. It can be determined from a cubic equation resulting from the derivative of (8). It is $a = 0.483$ for $\gamma_N = 0.5$ and $a = 0.453$ for $\gamma_N = 0.75$. If $\gamma_N =1$, the baseline correlation is $\gamma_G = 0.618$, and the phase noise $\sigma_{\Delta\phi} = 0.9$ rad. The former is in agreement with the "optimum correlation" derived by Rodriguez and Martin (1992) for $\gamma_N = 1$.

The expression $sin\theta \, cos\theta \, 2^{-3/2}\pi^{-1}$ ranges from 0 at $\theta$=0° and 90° to a maximum of 0.0563 at $\theta$=45°. In Fig. 1a, the
normalized relative height error $\sigma_h/\Delta y$ is shown as a function of $B_n/B_{cn}$ for $\gamma_N =1$ and incidence angles of 25° and 40°. Relatively small height errors can be obtained over a wider interval of $B_n/B_{cn}$ - ratios from 0.2 to 0.6. In Figure 1b, the effect of system noise is demonstrated for an incidence angle of 25 deg, assuming values of $\gamma_N = 0.75$ and 0.5, corresponding to low SNRs of 5 dB and 0 dB, with the latter as limiting case. A low SNR not only affects the achievable height accuracy and the





length of the optimal normal baseline but also narrows the interval of $B_n/B_{cn}$ – ratios in which the $\sigma_h$ can be regarded as still acceptable. Note that the ratio $B_n/B_{cn}$ does not depend on $p$.

## 3 Results

Besides the TanDEM-X mission, satellite configurations operated at other frequency bands are taken into account: (a) Tandem-L (Krieger et al., 2010) (b) a C- band tandem with wavelength and orbit altitude of Sentinel-1 (Torres et al., 2012) considering the possibility of adding a passive companion satellite for interferometric measurements; (c) a $K_u$-band tandem based on a concept presented by Lopez-Dekker et al. (2014) for a single platform meant for measuring ocean currents; and (d) a scenario for $K_a$-band that is adopted from a proposal for an ESA Earth Explorer mission (Börner et al., 2010).

Wavelengths and orbit altitudes listed in Table 1 are taken from the references given above. In this study, we selected incidence angles of 25 deg and 40 deg for all five mission scenarios. The slant range resolution $\Delta\rho$ ($\Delta y=\Delta\rho/sin\theta_l$, where $\theta_l$ is the local incidence angle) for TSX is 1.2 m for the High-Resolution Spotlight (SL) and single-polarized Stripmap (SM) imaging modes (TerraSAR-X Ground Segment, Basic Product Specification Document, http://sss.terrasar-x.dlr.de/). For Sentinel-1 Stripmap mode, the slant range resolution is 2 m at $\theta_l$=25.6 deg and 3.3 m at $\theta_l$= 41 deg (Aulard-Macler, 2012). For Tandem-L we used a bandwidth of 85 MHz (Krieger et al., 2010) for calculating $\Delta y$. Tandem-L orbit parameters are under discussion (status end 2016: altitude 745 km). For the $K_u$-band mission we assumed a bandwidth of 100 MHz instead of 10 MHz as used by Lopez-Dekker et al., 2014. The bandwidth for SIGNAL (Ka-band) is assumed to be 40 MHz, based on discussions on the mission concept. The normal baselines given in the table are $0.382 \times B_{cn}$, with the critical baseline from equation 4. Height of ambiguity and relative height error are calculated using equations 2 and 3, respectively. The small ambiguity heights for TanDEM-X, e. g., may require phase unwrapping if the actual ridge height corresponds to a phase difference > $2\pi$. In a fringe corresponding to 10 m height difference, a ridge of 2 m height extends only over one fifth of the fringe width, which may be not sufficient for the retrieval of the ice surface relief, in particular if the phase noise is high. The mean maximum heights of first-year ice ridges reported for different areas of the Arctic range from 1.1 m to 3.3 m (Strub-Klein and Sudom, 2012), and the average heights across the transverse section of a ridge are even smaller (see section 4). Except for SIGNAL, the achievable minimum relative height errors (under ideal conditions) are hence on the limit that is necessary for a meaningful retrieval of a rough sea ice surface topography. From the investigated configurations, the lowest error is achieved with TanDEM-X, the largest for SIGNAL. If the bandwidth of a Ka-band mission is increased, e. g. to 100 MHz, the relative height error is similar to the $K_u$-band values (0.49 and 0.42, respectively).

The values of Table 1 are valid if the correlation related to system noise, $\gamma_N$, is still close to 1, i. e. the signal-to-noise ratio of the measurements is larger than about 17 dB (corresponding to $\gamma_N$=0.98). However, the noise-equivalent $\sigma_0$ (NESZ) for satellite SARs is relatively high and hence the SNR for thin and smooth level ice low. For the satellite missions used as examples in this study, noise-equivalent-sigma-zero (NESZ) values are given in Table 2. Examples of radar backscattering coefficients $\sigma_0$ typical for different ice classes and conditions are listed in Table 3. Dierking (2013) compared C- and X-band images of sea ice in the Beaufort Sea and found that their intensity variations was highly correlated for level and deformed



ice except for nilas (thin ice) forming in an area of open water. In about 85 percent of the investigated cases, the X-band intensities varied between -19 dB and -8dB and were 3-5 dB higher than at C-band. This compares well with measurements reported by Tucker et al. (1991) (see also overview presented in Dierking, 2012). Publications of radar measurements over sea ice at $K_u$- and $K_a$-band are sparse. Dierking (2012) summarized the results of field and laboratory studies. In the

incidence angle range from 25° to 40°, values between -15 dB and +7 dB are reported at $K_u$-band, with nilas revealing the lowest and multi-year ice the largest intensities. Corresponding numbers for $K_a$-band are -12 dB and +10 dB.

The ranges of backscattering intensities and noise levels for the different radar systems considered here indicate that the SNR typically varies between 0 and 25 dB under real conditions. Smooth thin level ice (without frost flower coverage, snow crusts, and air inclusions in the ice volume) reveals relatively low backscattering intensities, hence SNRs are between 0 and

10 dB. For thicker level ice (surface roughness scales from millimeters to centimeters) and ice with a considerable fraction of air inclusions, the SNR ranges from 5 to 15 dB in most cases. Deformation structures in the ice cover (e. g. ridges, brash ice), imaged  at higher spatial resolutions below about 50 m, may reveal spots of very large intensities originating from ice blocks and fragments with their surface oriented normally towards the radar. Stronger multiple and volume scattering arises from piles of ice blocks. The difference between the backscattering coefficient and the NESZ will hence be roughly 15-25

dB. For the three given ice classes, the corresponding values of $\gamma_N$ are 0.5-0.91 (thin level ice), 0.76-0.96 (thicker level ice), and 0.96-0.99 (deformation structures).

In Table 4, the effect of low backscattering intensities on the achievable height accuracy is demonstrated, using SNR = 10 dB and 5 dB. For the former case one obtains $\gamma_N = 0.91$, $B_n/B_{cn} = 0.418$, $\gamma_G = 0.582$, and $\sigma_{\Delta\phi} = 1.13$ rad, for the latter case $\gamma_N = 0.75$, $B_n/B_{c\,n} = 0.454$, $\gamma_G = 0.546$, and $\sigma_{\Delta\phi} = 1.55$ rad. The given signal-to-noise ratios are typical for new ice and smooth

first-year level ice at lower radar frequencies. For SNR = 10 dB, the relative height error is larger by a factor between 1.1 and 1.2, for SNR = 5 dB by a factor from 1.4 to 1.5.

A distinct surface topography and height variations of one meter and more are usually observed in areas of first-year ice (thickness >0.3 m) and multi-year sea ice (thickness > 2 m). Considering the ridge height statistics provided by Strub-Klein and Sudom (2012) the relative height error $\sigma_h$ should optimally be less or equal 0.5 m for meaningful height retrievals.

Ridges and extended deformation zones reveal large backscattering intensities at C- and L-band (Dierking and Dall, 2007). Hence, the SNR is large, but on the other hand, the SNR over level ice areas is relatively low and the relative height error correspondingly large. If one, e.g., assumes that ice ridges, which appear as narrow high intensity zones in a SAR image, are distributed in areas of level ice with lower backscattering intensities, the SNR and hence the achievable height accuracy may vary significantly within short distances. This effect is less severe at higher radar frequencies (X-, $K_u$-, and $K_a$-band), since

the intensity contrasts between deformed and level ice are considerably lower. The reason is that the relative backscattering from level ice is stronger since its surface appears rougher to the shorter radar wavelengths. This means that the differences of the SNRs between deformed and level ice are not as large as at the lower frequencies, which is an advantage for topographic mapping.



## 4 Examples

The bistatic formation during the TanDEM-X Science Phase started on March 12, 2015. All in all, over 40 bistatic image pairs were acquired around Barrow, off the Coast of Alaska, USA, predominantly with large interferometric baselines. In the area of interest, a station is located that is equipped with sea-ice radar and a webcam, which both acquire imagery regularly.

For our study two TanDEM-X image pairs were selected to generate preliminary maps of sea ice surface topography by applying a standard SAR-interferometric approach. The main processing steps included spectral filtering of the input images, interferogram generation and flat earth removal, interferogram filtering, and phase-to-height conversion.

The first example shown in Fig. 2, was generated from data that were gathered on March 29, 2015, close to the coastline of Barrow. For the bistatic mode $p$ equals 1, the incidence angle $\theta$ was 27.3 deg, the normal baseline $B_n$ amounted to 1113 m,

and the length of the along-track baseline was 138 m. With an orbit height of $H = 514$ km and a radar wavelength $\lambda = 0.031$ m one obtains $h_a = 7.4$ m for the height of ambiguity (equation 2). The critical normal baseline $B_{cn}$ is 8072 m (equation 4), and the relative error $\sigma_h$ varies between 0.66 m for a signal-to-noise-ratio of SNR=10 and 0.51 for SNR=100 (equation 8, SNR given as linear value). The area from which the elevation profile depicted in Fig. 2c was retrieved was landfast ice, hence we can neglect contributions of ice movement to the interferometric phase caused by the along-track component of the

baseline (this issue is addressed in the next section). The profile reveals single prominent ridges with realistic heights, and rugged terrain between distances of 2400 and 4000 m from the selected origin. Unfortunately, coincident data of surface topography obtained by other sensors (e.g. laser profiler or scanner) are not available for this area and day. The general characteristic of the ice surface structure obtained from SAR compares well with the structures that can be recognized in the webcam image of the Barrow station and in the sea ice radar image. Both are shown in Fig. 3.

An empirical estimation of the relative height error is derived by evaluation of the local height statistics within a representative area with a relatively flat and homogeneous sea ice surface (red polygon in Fig. 2a). The area was located close to the coastline, several kilometers northeast of Barrow. The standard deviation of the surface height for this sample area is 0.12 m, calculated from the retrieved DEM, which has a spatial resolution of 12 m. The one-look resolution of the data was 2.5 m in ground range and 6.6 m in azimuth direction. Assuming that the standard deviation is caused by noise

effects and neglecting the correlation between adjacent pixels, the number of looks in the height map is approximately 8.7, and the theoretical relative height error is between $0.51/\sqrt{8.7}$ to $0.66/\sqrt{8.7}$ m, i. e 0.17 and 0.22 m. The empirical evaluation of a local height statistics hence compares reasonably with the theoretical derivation in section 2.

A second example from an area located southwest of Barrow can be found in Figure 4. The data were acquired on March 20, 2015 with a normal baseline of 833 m, an along-track baseline of 42 m, and an incidence angle of 37.2 deg. The height

of ambiguity is 14.5 m. The amplitude image (Fig. 4b) reveals that the profile - when starting at point B and moving to the left - crosses a fast ice zone (dark grey belt with bright structures), an open water area with indications of wind-driven Langmuir circulation (dark grey area with bright stripes), possibly a narrow zone of thin ice (dark grey zone), and pack ice (bright grey) in which open water leads (dark areas) are embedded. In the height map, the open water area (distance from





6000 to 10000 m) and the lead (400 to 1500 m) crossed by the profile appear as rugged ice terrain in the height map with heights between two and almost eight meters. This example demonstrates the strong influence of surface motions in the radar line-of-sight direction, which adds a non-negligible height offset to the final topographic map. The along-track baseline of 42 m corresponds to a temporal baseline of 6 milliseconds. This time interval is shorter than the decorrelation time of a water

surface, which ranges from about 8 to 10 milliseconds at X-band (Romeiser and Thompson, 2000). Hence, the requirement for a measurable phase difference is fulfilled. The interferometric phase of open water areas is proportional to the mean surface current parallel to the radar look direction (in this case possibly induced by the Langmuir circulation) and contains also contributions associated with the velocity of small wind-induced ripple waves and with the surface currents due to the orbital motion generated by longer ocean waves (Romeiser and Thompson, 2000). The large "height" variations in the open

water areas of Fig. 4 may be caused by the alternating water and frazil ice stripes. Since at longer temporal baselines the interferometric coherence measured over water is very low, it can be used to detect the presence of openings in the sea ice cover imaged by radar. Not only areas of open water contaminate the height retrieval from InSAR measurements, but also the drift and rotation of the pack ice. This is discussed in detail in the following section.

## 5 Discussion

### 5.1 Influence of sea ice motion

Since most parts of the sea ice cover are in steady motion, along-track baselines cause additional phase shifts that affect the retrieval of topographic heights (see equation 1). In addition, the movement of the ice between the acquisitions of image 1 and image 2 leads to decorrelation effects due to speckle. In case of single-pass InSAR with very small along-track baselines (otherwise surface height retrievals will be severely hampered, see below), the effect of temporal decorrelation can

be neglected for sea ice. The interferometric phase $\Delta\phi_{mov}$ for a baseline $B_{al}$ in along-track direction (corresponding to the along-track distance of the positions from which the images are acquired) is (Madsen & Zebker, 1998):

$$\Delta\phi_{mov} = -\frac{2\pi p u_{LOS} B_{al}}{v\lambda},\qquad(9)$$

with $v$ – ground velocity, $u_{LOS}$ – line-of-sight or radial object velocity, $\lambda$ – radar wavelength, $B_{al}$ – along-track baseline, and $p$ is explained above. The radial velocity can assume positive and negative values, dependent on the direction of the movement

(towards or away from the radar). It is determined from the sea ice drift velocity $u$ by $u_{LOS} = u\,sin\theta\,cos\varphi$, with $\theta$ - incidence angle, and $\varphi$ - azimuth angle between the direction of the ice drift and its across-track component. Here it is assumed that the vertical component of the ice displacement is zero. Whether this assumption is justified is discussed below. For a given phase difference the along-track baseline depends linearly on $\lambda$ and $v$, and decreases with increasing $u_{LOS}$. Typical average sea ice drift velocities range mostly from 0 to 0.35 m/s (1.26 km/h)  (Rampal et al., 2009), but for instantaneous radial

velocities Kræmer et al. (2015) found values up to 6 m/s from analyses of the Doppler shift of SAR signals.





For TanDEM-X, the ground velocity is 7 km/s. In the following discussion we determine the along-track baseline which at a given line-of-sight-velocity causes a phase shift corresponding to a given relative height error.  If the height error is set to $\sigma_h = 0.5$ m (which represents a still acceptable accuracy, see above) and the height of ambiguity to 5 m (representing one fringe), the corresponding phase difference amounts to 36° or $0.2\pi$ rad. With $p=2$ and $\lambda = 0.031$ m, a phase shift $\Delta\phi_{mov} = 0.2\pi$

rad is obtained at $B_{al} = 339$ m (56 m), if $u=0.05$ m/s (0.3 m/s), $\varphi = 0°$, and $\theta = 40°$, which gives $u_{LOS} = 0.032$ m/s (0.193 m/s). The baseline length $B_{al}$ doubles if $p = 1$. In units of time, the temporal baselines are $B_{al}/v = 0.05$ s and 0.008 s, hence extremely short. Also here the phase noise has to be considered, which gives a relative velocity error of $\sigma_{uLOS} = v\lambda\sigma_{\Delta\Phi} / (2\pi p B_{al})$. If the SNR is low and the baseline decorrelation not negligible, critical phase differences due to surface motion are reached at even shorter baselines. In the examples presented below, we assume that the SNR $\geq 15$ dB.

In Figure 5, the "critical system-normalized" along-track baseline $B_{aln} = | pB_{al} / v\lambda |$ is plotted as a function of the magnitude of the line-of-sight velocity for different ratios $\Delta\phi_{mov} /2\pi$. The latter are calculated from $\sigma_h /h_a$ (equation 3). The notation "critical" means to regard this baseline length as the threshold for which the phase shift corresponds to a height change of 0.5 m.  For line-of-sight velocities larger than 0.2 m/s, the change of the system-normalized baseline is only small. For TanDEM-X data, e. g., $B_{aln}$ has to be multiplied by $v\lambda/p = 217/p$ [m²/s]. For $u_{LOS} = 0.032$ m/s and a ratio $\phi_{al} /2\pi = 0.1$ one

obtains $B_{aln} = 3.125$ s km⁻¹, i. e. $B_{al} = 678$ m if $p = 1$. If the sea ice drifts in across-track direction, the corresponding velocity is 0.075 m/s at $\theta = 25°$ and 0.05 m/s at $\theta = 40°$. In Table 3, "worst case" critical along track baselines are listed (for $p = 1$) that cause phase shifts corresponding to a relative height error of 0.5 m at an ambiguity height of 5 m. For $p = 2$, they are even shorter by a factor of two. Table 3 reveals that extreme instantaneous line-of-sight velocities, such as reported by Kræmer et al. (2015) cause significant phase shifts already at very short baselines, in particular at higher radar frequencies.

The sensitivity to ice motion even increases at larger height ambiguities (with $\sigma_h$ fixed at 0.5 m), as is demonstrated in Fig. 5. Since according to equation 1, surface topography and movement affect the measured interferometric phase simultaneously, it is necessary to obtain independent data of the line-of-sight velocity. This can in principle be achieved by estimating the Doppler centroid from the unfocused SAR data as described in Kræmer et al. (2015). For this method, however, a sufficient number of neighboring pixels has to be averaged, resulting in spatial resolutions on the order of hundreds of meters to a few

kilometers. Whether this approach is feasible in practice needs to be investigated in detail in another study.

So far, it was assumed that the motion of the ice is rectilinear, i. e. along a straight line, during the time interval between the acquisitions of images 1 and 2. However, also rotational motion of single ice floes about their vertical axis causes phase shifts and leads to an additional decorrelation effect (Zebker and Villasenor, 1992; Scheiber et al., 2011). The maximum magnitudes of floe rotation rates vary between 0.02 deg per hour in the central Arctic with its compact ice cover and 2 deg

per hour in the marginal ice zone, where the ice concentration is low (Leppäranta, 2011). In the latter, rates of even more than 100 deg per day were noted at rare occasions which may have been caused by ocean eddies. Considering the temporal baselines given in Table 5, the expected rotation angles that occur during single-pass along-track InSAR data takes vary between $10^{-9}$ deg  and $10^{-3}$ deg. From interferograms derived from TanDEM-X ScanSAR images acquired at the NE coast of Greenland, Scheiber et al. (2011) retrieved floe rotations up to 0.005 deg for a time interval of 2.6 s between the two image





acquisitions. This demonstrates the very high sensitivity to rotational movements, which is also valid for rotations around a horizontal axis. The rotational phase shift depends on the azimuth position $x$ relative to the center of rotation and the rotation angle $\varphi_{rot}$:

$$\Delta\phi_{rot} = \frac{4\pi}{\lambda} x \ sin(\varphi_{rot})\sin(\theta) \tag{10}$$

where $\lambda$ and $\theta$ are explained above. The rotation angle is measured relative to the azimuth direction (Scheiber et al., 2011). Due to the rotation, the scattering elements in a resolution cell change their position, which causes decorrelation of the received radar signals. Total decorrelation occurs at a rotation angle of $\lambda/(2\ \Delta x\ sin\theta)$, where $\Delta x$ is the azimuth resolution, i. e. at higher radar frequencies and larger incidence angles the "critical" rotation angle is smaller. For TanDEM-X ScanSAR and Stripmap mode, the respective angles are ±0.086 deg and ±0.25 deg for θ=30 deg (Scheiber et al., 2011).

In the analysis above it was assumed that the vertical ice motion is zero. As Mahoney et al. (2016) demonstrated, already small vertical displacements of a few millimeters (as observed, e. g., when infra-gravity waves propagate in sea ice covered areas) may cause significant phase shifts. However, in their investigation the temporal baseline was 10 s. They reported wave amplitudes between 1.2 mm and 1.8 mm with periods between 30 s to 50 s. For topographic mapping, very short temporal baselines are required, at X-band, e. g., optimally less than 0.5 s and even much less if the line-of-sight velocities

are high (see Table 5). Hence, vertical displacements caused by infra-gravity waves can be neglected. However, a sudden deformation event due to pressure or shear forces in the ice, resulting in a vertical shift of smaller ice areas, may cause non-negligible phase shifts. But since such process is a momentary event, the probability that the related movement is directly measured is very low.

### 5.2 Influence of penetration depth and horizontal resolution

Another important question that needs to be investigated is whether the sea ice surface height retrieved from the interferometric data represents the actual height. Unfortunately, this is not the case. One has to consider two effects: (a) Over rugged sea ice terrain, the retrieved value is an effective height determined by the spatial resolution of the interferogram, and (b) the radar waves penetrate into the ice and snow, which means that the horizon of maximum backscatter does not necessarily correspond to the true ice (or snow) surface. In addition, the effect of volume decorrelation has to be considered

(see below).

Strub-Klein and Sudom (2012) present numbers for the maximum height of ridge sails and for average sail heights (in both cases they list values for maximum, minimum and mean). The average sail height is the mean of the heights measured over a ridge cross section. Considering typical horizontal resolutions of topographic maps derived from InSAR data, the retrieved interferometric height will closely correspond to the average sail height. To be more specific, the individual widths of ridge

sails reported by Strub-Klein and Sudom (2012) range from 1.8 m to 73.2 m with mean values between 9.6 m and 17.5 m for different locations in the Arctic and 7.4 m for the Baltic Sea. The corresponding average sail heights were between 0.1 m and 3.3 m (minimum and maximum from all individual ridges), with mean values from 0.3 m to 1.6 m for the different Arctic





locations and 0.3 m for the Baltic. If the sail width is larger (smaller) than the spatial resolution of the interferogram, the retrieved height will be larger (smaller) than the average sail height as defined above. Since ridges of low height reveal small widths, the height error of the interferometric retrieval may be too large to determine any useful value. Referring to Fig. 6 in the article by Strub-Klein and Sudom (2012), most ridges with widths > 10 m are between 1.5 meter and 8 m high

(maximum values). The statistics presented by Strub-Klein and Sudom (2012) are based on measurements of first-year ice ridges. Multi-year ridges (i. e. ridges which survived at least one melting period) are more rounded, and the degree of consolidation (bonding between single ice blocks) is higher. Because of the lack of a multi-year ice ridge statistics we assume that their ridge height distribution is similar, and their average width is larger.

Freeboard values retrieved from radar altimeter data are typically lower than 0.5 m, only north of Greenland's coast, higher

values may occur (Ricker et al., 2014). With the relative height errors listed in Table 1, the estimation of ice freeboard at the edges of leads is at the limit of the achievable relative accuracy.

The penetration depth $d$ of radar waves (in terms of power) into ice depends on ice salinity, temperature, and radar frequency. Note that we refer to the one-way penetration depth, which is $d = \kappa^{-1}$, if the extinction coefficient $\kappa$ is constant with depth. It depends both on the real and imaginary part of the dielectric constant. If the latter is close to zero, the

penetration depth approaches infinity. For saline first-year ice, $d$ decreases if the ice temperature and salinity increase. Since the salinity of multi-year ice is low, variations of the penetration depth are dominated by temperature changes. Under freezing conditions and if volume scattering is negligible, the penetration depths at X-band are about 1-7 cm into first-year ice and 5-30 cm into multi-year ice, which is of lower salinity. The corresponding values for C-band are roughly twice as large. For $K_u$-band ($K_a$-band), penetration depths range between 3 and 17cm (2-8cm) for multi-year ice, and between 0.5 and

5 cm (0.3-2cm) for first-year ice. These numbers were taken from Lewis et al. (1987). At L-band, the penetration depths are 0.3 – 1 m for first-year ice and 1 – 3 m for multi-year ice (Ulaby et al., 1986, Appendix E). In Shokr and Sinha (2015, Table 8.11), the following values are given for first-year ice with a snow cover of 13 cm and multi-year ice with 20 cm overlaid snow: L-band 49/160cm, C-band: 7.0/32.0cm, X-band: 4.0/20.0cm, $K_u$-band: 3.3/18.4cm (interpolated value), $K_a$-band 1.0/9.0cm. Since ice blocks in ridges are often desalinated, the effective penetration depth into the ridged ice is larger than

into the adjacent level ice, which reduces the apparent ridge height relative to the level ice surface retrieved from the interferogram. In the following we quantify the effect of the penetration depth.

**5.3 Volume decorrelation**

In section 1, we defined the interferometric coherence as $\gamma = \gamma_G \gamma_N$, assuming that volume scattering can be neglected. In case of low-salinity sea ice, this is not always the case, which means that the coherence includes a volume component: $\gamma = $

$\gamma_G \gamma_N \gamma_{Vol}$. The effect of volume decorrelation can be estimated based on equation (11), which was derived by Weber Hoen and Zebker (2000):

$$|\gamma_{Vol}| = \frac{1}{\sqrt{1+\left(\frac{\pi p B_n \cos\theta}{H \lambda \tan\theta}\sqrt{\epsilon'}d'\right)^2}} \qquad (11)$$





with $B_n$, $p$, $H$, and $\theta$ defined above. Here, $|\gamma_{Vol}|$ is the correlation coefficient, $\varepsilon'$ is the real part of the dielectric constant of the ice, $d'$ is the penetration length along the propagation direction of the refracted wave at which the one-way power falls to $e^{-1}$, and $\lambda$ is the radar wavelength in free space. Note that Weber Hoen and Zeber derived equation 10 for the case of repeat-pass intereferometry, i. e. $p = 2$. The penetration depth $d$ (along the vertical) is $d = d' \cos\theta_r$, where $\theta_r$ is the refraction angle. The

correlation decreases if the penetration depth into the ice increases. In the following discussion, a radar resolution cell corresponds to a volume element. Equation (11) is derived under the assumptions that (a) the scattering medium is homogeneously lossy, (b) the radar cross section of the scattering elements varies only as a function of depth, (c) the volume is characterized by an exponential extinction, and (d) the layers at depths $> d$ do not contribute to the backscattered signal. In the derivation a non-weighted, ideal radar transfer function is used (Weber Hoen and Zebker, 2000). Based on the study by

Dall (2007), we modified equation 11. From Snell's law we obtain $\cos\theta_r = (1 - \sin^2\theta /\varepsilon')^{-1/2}$. If the radar waves penetrate into the volume, the height of ambiguity changes according to

$$h_{a\_Vol} = h_a \frac{\sqrt{\varepsilon' - \sin^2\theta}}{\varepsilon' \cos\theta} = h_a \, c_{\varepsilon\theta} \qquad (12)$$

(Note that there is a printing error in this equation in the paper by Dall, 2007). Equation 11 then simplifies to

$$|\gamma_{Vol}| = \frac{1}{\sqrt{1 + \left(\frac{\pi d}{h_{a\_Vol}}\right)^2}} \qquad (13)$$

Equation 13 represents the absolute value of equation 9 in Dall (2007), except that Dall uses the two-way penetration depth $d_2 = d/2$. Note that $h_a$ and hence $|\gamma_{Vol}|$ are functions of $p$. According to equation 13, the volume correlation depends on the ratio between the penetration depth and the volume-corrected height of ambiguity. However, equation 13 is only valid if the ice thickness exceeds the penetration depth by a factor of 2-5, otherwise the volume correlation additionally depends on the ratios $D/h_{a\_Vol}$ and $D/d$ (Dall, 2007). The correlation coefficient as a function of the ratio $d/h_{a\_Vol}$ is shown in Fig. 6. For the

dielectric constant of sea ice, results of measurements are presented in Hallikainen and Winebrenner (1992, their Figs. 3.5 and 3.6) for different ice types, dependent on salinity and temperature. Those measurements were carried out in the frequency ranges 4 – 5 GHz and 10 - 16 GHz for salinities between 0.2 ppt and 0.5 ppt and temperatures between -50°C and -0.2°C. In the first frequency interval, the real part of the dielectric constant assumes values between 2.9 and 4.3, in the second one between 2.5 and 4.2. Values for multi-year ice are between 2.5 and 3.1, for first-year ice between 2.9 and 4.2.

For the discussion of examples presented in Table 6, we assume $\varepsilon' = 2.8$ for multi-year ice and $\varepsilon' = 3.5$ for first-year ice, yielding $c_{2.8,25} = 0.6380$, $c_{3.5,25} = 0.5745$, $c_{2.8,40} = 0.7203$, $c_{3.5,40} = 0.6553$ for the coefficient $c_{\varepsilon\theta}$ in equation (11). A value of $d/h_{a\_Vol} = 0.1$ corresponds to $|\gamma_{Vol}| \approx 0.95$.

If the finite thickness of sea ice is taken into account, the volume decorrelation is a function of the three parameters ambiguity height $h_{a\_Vol}$ (which characterizes the radar system), penetration depth $d$ (which depends on salinity, temperatue,

and the fraction, size, and shape of air bubbles in the ice), and ice thickness $D$. The radar waves do not penetrate into the





saline water below the ice. Hence we can apply equation 8 given in the paper by Dall (2007). Evaluating the magnitude of his expression, we obtain

$$|\gamma_{Vol}| = \frac{\sqrt{\left[1-A\exp\left(-\frac{2D}{d}\right)\right]^2 + \left(\frac{\pi d}{h_{a\_Vol}}\right)^2\left[-1+B\exp\left(-\frac{2D}{d}\right)\right]^2}}{\left[1+\left(\frac{\pi d}{h_{a\_Vol}}\right)^2\right]\left[1-\exp\left(-\frac{2D}{d}\right)\right]} \qquad (14)$$

with

$A = \cos\frac{2\pi D}{h_{a\_Vol}} - \frac{\pi d}{h_{a\_Vol}}\sin\frac{2\pi D}{h_{a\_Vol}}, \qquad B = \cos\frac{2\pi D}{h_{a\_Vol}} + \frac{h_{a\_Vol}}{\pi d}\sin\frac{2\pi D}{h_{a\_Vol}}$

The coefficients $A$ and $B$ give rise to a resonance phenomenon that owing to the multiplication with *exp(-2D/d)* only occurs when the penentration depth is larger or of same magnitude as the ice thickness. Note that scattering from the ice-water interface is not considered here. In the case of low ice salinity, measurable scattering contributions from the ice-water boundary may occur as was demonstrated by Dierking et al. (1999) for Baltic Sea ice. The development of a model including

the interface scattering, however, is beyond the scope of this study.

The comparison of the critical penetration limits and the penetration depths listed in Table 6 reveals that volume decorrelation can be neglected at $K_a$- and $K_u$-band both for first- and multi-year ice. Equation 13 is not applicable for thin young ice (thickness < 5 cm) but in this case topographic undulations can not be reliably retrieved considering the achievable height accuracies. At X-band, volume decorrelation has to be taken into account for low-salinity multi-year ice, and equation

13 is still applicable for a larger range of the ice thickness. For the Ka-, Ku-, and X-band mission scenarios shown in Tables 1 and 6, the ratio $d/h_{a\_Vol} \ll 1$. This means that according to Dall (2007) the elevation bias (relative height error) due to volume effects equals half the one-way penetration depth. If we focus on first-year ice with $D > 0.5$ m (note that even on ice with $D \approx 0.2$ m, ridge sails may be as high as 3 m, see Fig. 15 in Strub-Klein and Sudom, 2012), and assume that the salinity of thinner second-year ice is larger than for elder multi-year ice (which means a smaller penetration depth in the first case),

useful estimates of the critcial penetration depth according to equation 12 can still be obtained at C-band. The ratio $d/h_{a\_Vol}$ is $\ll 1$, which means that also in this case, the elevation bias approximately corresponds to 0.5$d$, that is at maximum to about 0.3 m for low-salinity multi-year ice (see penetration depths listed above). The situation at L-band is more complicated, since the ice thickness and penetration depths are of similar magnitude. Hence, equation 14 is applied to provide estimates for the critical penetration depths, which reveal that volume decorrelation has to be considered at L-band. The elevation bias

depends both on thickness and penetration depth. The derivation of a corresponding relationship is not carried out here. Dall (2007) only considers cases with $d/D$ and $h_{a\_Vol}/D$ approaching infinity for which he obtains a bias of 0.5$D$.

The determination of volume decorrelation and elevation bias requires that the penetration depth and hence ice salinity, temperature (and – of minor importance - volume fraction of scattering elements) have to be obtained parallel to the SAR data acquisitions which is not possible in practise. Optimal measurements conditions with relatively smaller penetration

depths are given if the ice temperature is close to the freezing point, but still too low for melt-onset. Classification maps





separating multi-year, first-year, and thin ice obtained from the SAR intensity images are helpful for judging the reliability of the estimated height error. The separability of ice classes, however, depends on radar frequency and polarization. We note that the application of equations 13 and 14 requires that volume scattering is not negligible (see Dall, 2004, equation 2). Volume scattering may be very low under certain conditions, in particular at L-band. The presence of snow on the ice (see

below) complicates the situation further. Tomographic radar measurents on sea ice such as reported by Yitayew et al. (2016) may provide useful insights regarding these issues.

### 5.4 Influence of snow

In real-world situations, snow layers are present on the ice. Dry snow is almost transparent at larger radar wavelengths (penetration depth, e. g. 30 m at a wavelength of 7.5 cm) and still highly penetrable at smaller wavelengths (1.5 m at 1 cm),

see Ulaby et al. (1982, Fig. 11.25). Scattering from the snow surface is negligible in most cases. In the snow, the radar wavelength decreases, and the incidence angle at the snow-ice interface is smaller than at the air-ice interface. Hence, the results given above for a snow-free ice surface have to be adjusted accordingly. Effects are, e. g. that ambiguity height and critical baseline decrease (equations 2 and 4). In general, snow thickness is larger in areas of deformed ice. From measurements at different sites in the Arctic, Sturm et al. (2006), e. g., reported snow thickness variations between a few

centimeters and up to 80 cm with mean values between 9 and 21 cm. The snow thickness may vary considerably on relatively short spatial scales, e. g. due to redistribution by wind. In the intervening smooth ice, e. g., the snow layer may be thicker than on top of the ridges but less thick than at their lee sides. The snow density, which determines the dielectric constant of dry snow, may also vary. If snow thickness and density over the ridge and the adjacent level ice are different, the total topographic phase difference includes a contribution from the different paths along which the radar waves propagate.

If a homogeneous layer of snow with thickness h is assumed, the difference of the paths without ($s$) and with snow ($s'$) are $\Delta s = s - s' = h\,(cos\theta^{-1} - cos\theta_r^{-1})$. The refraction angle is $sin\theta_r = sin\theta\,\varepsilon'_{ds}{}^{-1/2}$. The real part of the dielectric constant for snow is related to its density $\rho_{ds}$ by $\varepsilon'_{ds} = 1 + 1.9\rho_{ds}$ for $\rho_{ds} \leq 0.5$ g/cm$^3$ and $0.51 + 2.88\rho_{ds}$ for $\rho_{ds} > 0.5$ g/cm$^3$ (Hallikainen and Winebrenner, 1992). The normalized path difference $\Delta s/h$ is shown in Fig. 7. If, e. g. the ridge is snow-free and the snow layer on the neighboring level ice with $\rho_{ds} = 0.6$ g/cm$^3$ is 40 cm thick, one obtains differences of 1.5 cm, 3.7 cm, and 11.2 cm

at incidence angles of 20°, 30° and 45°, respectively. Considering that these values are considerably smaller than the "acceptabe" relative height error of 0.5 m, the influence of a dry snow layer can be neglected in most cases. This is a valuable result since snow density and thickness data valid for the time of SAR image acquisitions are usually not available.

In case of strong surface winds and longer temporal baselines, the snow drift may reduce the interferometric coherence. This effect is more pronounced at higher radar frequencies. At K$_u$- and K$_a$-band, the major scattering horizon may not be

identical with the snow-ice surface but be located higher up in the snow layer (e. g. Willat et al. 2010, Willat et al., 2011). However, taking into account a realistic height error for retrievals from InSAR measurements, the rise of the scattering horizon is negligible. In moist snow, penetration depths decrease significantly. If the volume moisture content is 1 percent (5 percent), the depths for the given wavelengths are 70 cm (20 cm) and 5cm (1 cm) (Ulaby et al., 1982, Fig. 11.25.) During the





melting season, the radar signal is backscattered from the wet snow or – if no snow is present – from the wet ice surface. One could argue that height retrievals from images measured over melting ice provide the "real" surface that determines the aerodynamic drag. However, topographic data are also required from the winter season, and temporal variations and trends of the ice surface height need to be known for estimating the ice mass balance. It has also to be considered that the

backscattered intensity changes seasonally. E. g. in the case of multi-year ice, the volume scattering contribution is suppressed under melting conditions, hence the total backscattering decreases. The backscattered intensity may increase if superimposed ice is formed on top of first-year ice, but it may also decrease if melt water smoothens the small-scale surface roughness (Onstott et al., 1987).

### 5.5 Achievable baselines

InSAR techniques can successfully be applied for drifting sea ice only if image pairs are acquired with small temporal gaps on the order of milliseconds to seconds and baselines smaller than the critical limit determined by equation (4). This means that data from satellite configurations such as TanDEM-X are required. The two satellites of the TanDEM-X mission ("TerraSAR-X" and "TanDEM-X") fly in a helix-formation, which combines an out-of-plane (horizontal) orbital displacement due to different ascending nodes with a radial (vertical) separation due to different eccentricity vectors (Krieger

et al., 2007). The ascending node is the intersection of the equatorial plane and the satellite orbit on the leg from the southern to the northern hemisphere. The TanDEM-X satellite is controlled with respect to TerraSAR-X (Maurer et al., 2016). The maximum baseline varies along the orbit, its length is expressed as a function of the geographical latitude (AO TanDEM-X Science Phase manual, https://tandemx-science.dlr.de/). Furthermore, the effective baseline is larger at smaller (steeper) incidence angles. During the TanDEM-X Science Phase, the largest cross-track baselines amounted to 3000 m and were

achieved over the equator (AO TanDEM-X Science Phase manual, https://tandemx-science.dlr.de/). For a given latitude, the baseline length can be changed by varying the eccentricity vector of the orbit. The helix parameters are usually kept constant for certain periods to minimize fuel consumption. For a satellite tandem we can conclude: (a) It is in principle possible to achieve the cross-track baselines necessary for mapping height variations of the sea ice surface, (b) the sensitivity to surface height variations is not constant but varies as a function of latitude, and (c) optimal conditions for measurements

of the sea ice surface topography in a given region are only possible during limited temporal intervals because of satellite operation requirements. An important issue is the magnitude of the along-track baseline $B_{al}$ as we discussed above. For TanDEM-X, the uncertainty of estimates of $B_{al}$ amounts to ±200 m. As a rule of thumb, $|B_{al}|$ is twice as large as $B_n$ in a bistatic configuration. For TanDEM-X, it is at its maximum at the equator and approaches zero at the poles (Krieger, personal communication, December 2016).

In the literature, satellite constellations consisting of more than two receiver microsatellites (and a satellite with an active SAR ahead or behind) have been discussed (e. g. Krieger et al., 2003; Moreira et al., 2002). The general advantage of configurations consisting of N>2 receiver satellites is that the variations of the across-track baseline lengths as a function of latitude can be minimized by picking the most suitable transmitter – receiver combination. The interferometric cartwheel, e.





g., consist of satellites flying in close formation on slightly different elliptical orbits. The orbit parameters are selected such that the formation of receiver satellites seem to move on an ellipse centered on the orbit of the active satellite. However, with the cartwheel, across- and along-track baselines cannot be optimized at the same time. This is a disadvantage for retrieving the surface relief of drifting sea ice (see above). An alternative is the cross-track pendulum (with the TanDEM-X helix as a

special case). Here, the receiver satellites are all moving with equal velocities along circular orbits in different orbital planes with slightly different ascending nodes and/or inclinations. With this configuration, across-track baselines of any desired length can be formed. If three receiver satellites are used and the respective maximum baseline is selected, the variations are limited between 87% and 100% of the achievable maximum. At the same time, the along-track baselines can be set independently and kept constant (Krieger et al., 2003, Fig. 2). However, very short along-track baselines increase the risk of

collisions at crossing points of the orbit planes. Because of the secular drift of the ascending nodes (due to the nonspherical shape of the Earth) and the different inclinations, the cross-track pendulum formation is not stable. For maintaining the orbits, additional fuel is required. It is beyond the scope of this study to propose an optimal satellite formation for the retrieval of sea surface height undulations. But it is noted that this is a necessary requirement for planning future satellite missions suitable for determining surface topography on meter and sub-meter scale.

**5.6 Other factors**

When estimating the achievable accuracies of height retrievals, the effects of retrievals in the determination of (a) length and angle of the normal baseline, (b) local incidence angle, and (c) orbit altitude have to be considered and assessed routinely in InSAR processing. This is not made subject of this study, since the intention was to discuss specific conditions related to the retrieval of the topography of fast and drifting sea ice. Other factors that need to be taken into account in

InSAR processing are the accuracies of co-registration of the two images used for generating the interferogram, filtering steps for reducing the phase noise, flat-plane phase removal, and phase unwrapping (Richards, 2007). Phase unwrapping, however, may only be required for large sea ice ridges and low heights of ambiguity. The simplest approach for phase noise reduction in the interferograms, e. g., is achieved by averaging neighboring pixels, thus increasing the number of looks, $N_L$, which reduces the relative height error (see equations 3 and 5) but at the same time worsens the spatial resolution which

possibly decreases the retrieved apparent ridge heights.

**6 Conclusions**

In this paper we analyzed the application of interferometric SAR for retrieving the surface topography of sea ice, assuming different satellite missions with radar frequencies ranging from $K_a$- to L-band. As a basis for judging the feasibility we used statistics of ridge heights and widths reported in the literature. Optimal across-track baselines for achieving the lowest

possible height error vary from 40 km at L-band (incidence angle 40 deg) to 320 m at $K_a$-band (at 25 deg). Relative height errors smaller than 0.5 m are achievable for large signal-to-noise ratios (SNR>15 dB). In particular undeformed thin ice (without frost flower coverage) and smooth level ice reveal a low SNR. For an SNR of 10 dB, the relative height error


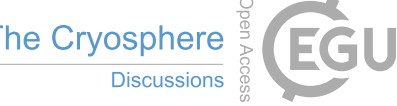

increases by a factor of 1.1-1.2, for SNR=5 dB the factor is 1.4-1.5. In case of drifting pack ice, the influence of the ice motion on the interferometric phase must be considered unless the line-of-sight ice velocity $u_{LOS}$ equals zero. For $u_{LOS}$ = 0.18 km/h, along-track baselines from 3400 m at L-band to 110 m at $K_a$-band cause phase shifts corresponding to a relative height error of 0.5 m. If $u_{LOS}$ = 2.2 km/h, which represents large wind speeds, the respective numbers are 280 to 10 m. Wind-driven

surface currents on open water areas within the ice cover may also generate a phase shift. Hence, such areas should be masked in the topographic map. Effects of volume scattering in ice and snow are negligible at $K_a$ and $K_u$-band and of minor importance at X-band because the radar penetration depths are relatively small at these frequencies. At C- and L-band, an increase of the height error due to volume decorrelation has to be considered in particular for low-salinity ice with large penetration depths. If a dry snow layer is present on the ice, the radar wavelength at the snow-ice interface is shorter than in

air, and the incidence angle steeper, changing the magnitude of the optimal across-track baseline. In case of melting conditions, radar penetration depths into the snow are reduced and approach zero at larger snow moisture content. With the recent TanDEM-X mission, a change of the default orbital parameters is required to achieve the necessary across-track baselines over the Polar Regions. The cross-track pendulum satellite configuration with more than two satellites can be more easily optimized for measurements of sea ice topography than the cartwheel. The availability of additional information in the

process of retrieving sea ice topography would be of advantage. For example, to judge the influence of sea ice motion on the height retrieval, the line-of-sight velocity should be determined simultaneously with the interferometric phase from the Dopplershift of the radar signal caused by the ice movement. Another valuable information is an ice chart showing the spatial distribution of different ice types, derived from the SAR intensity images used for generating the interferogram, possibly extended by images acquired at a different polarization and/or frequency with a satellite train configuration.

**Acknowledgement**

The authors express their thankfulness to our colleagues Jørgen Dall from the Technical University of Denmark and Gerhard Krieger from the German Aerospace Center for useful discussions and hints on volume decorrelation and orbit configurations. The TanDEM-X data were provided by the German Aerospace Center (DLR) under a scientific license, project: XTI_OCEA6971.





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





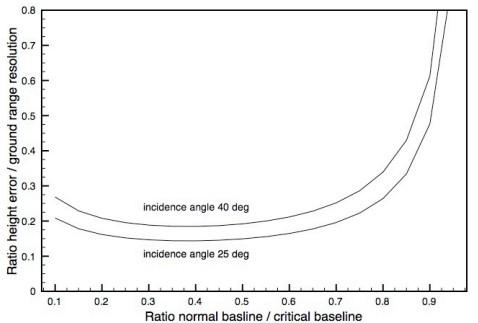
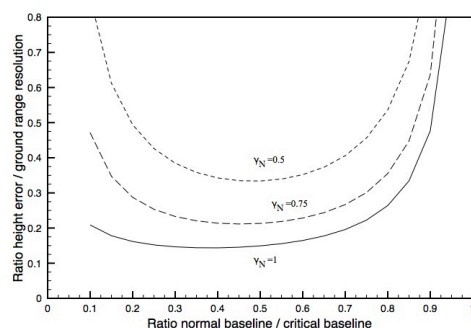

**Figure 1:** Relative height error normalized by ground range resolution as a function of the ratio between normal and critical baseline, shown for (a) $\gamma_N$ =1 and two incidence angles, and (b) for an incidence angle of 25 deg and different values of noise levels.



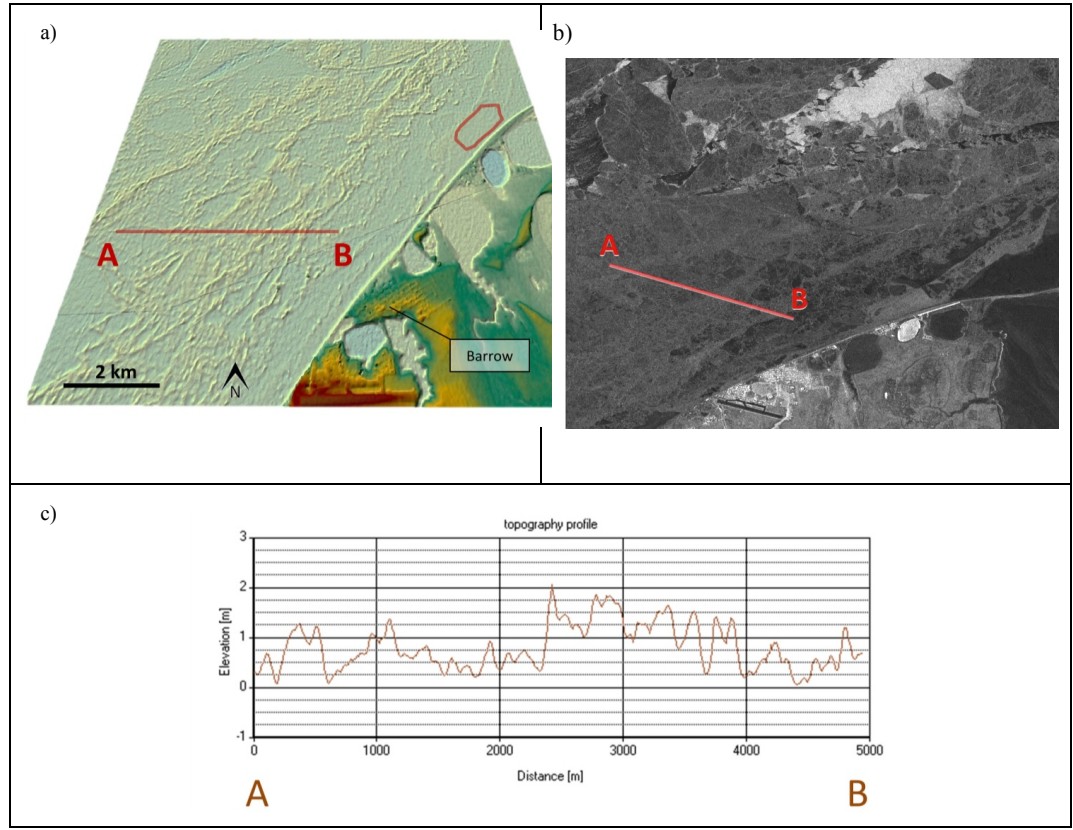

**Figure 2:** (a) Shaded surface topography map close to the coastline of Pt. Barrow, derived from data acquired on March 29, 2015. Shown is a subset of the full satellite scene. The sampling distance is 12 m. Red polygon: area for estimating the
5   height error (see text). (b) Corresponding TanDEM-X amplitude image (slant-range geometry). Line AB indicates the location of the surface topography profile in (c) © DLR e.V. 2015 and © Airbus DS Geo GmbH 2015





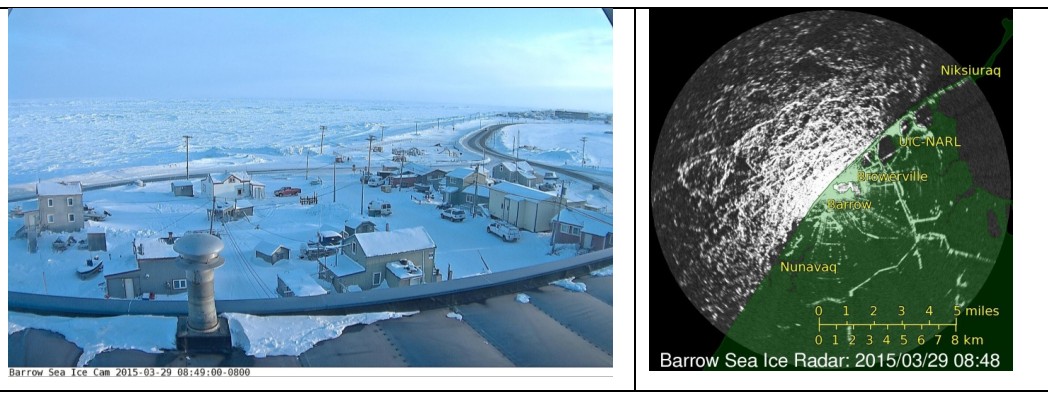

5   **Figure 3:** Webcam (left) and sea-ice radar image (right) obtained from the Geographic Information Network of Alaska,
University of Alaska Fairbanks, taken March 29, 2015 (http://feeder.gina.alaska.edu/radar-uaf-barrow-seaice-images)

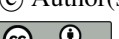



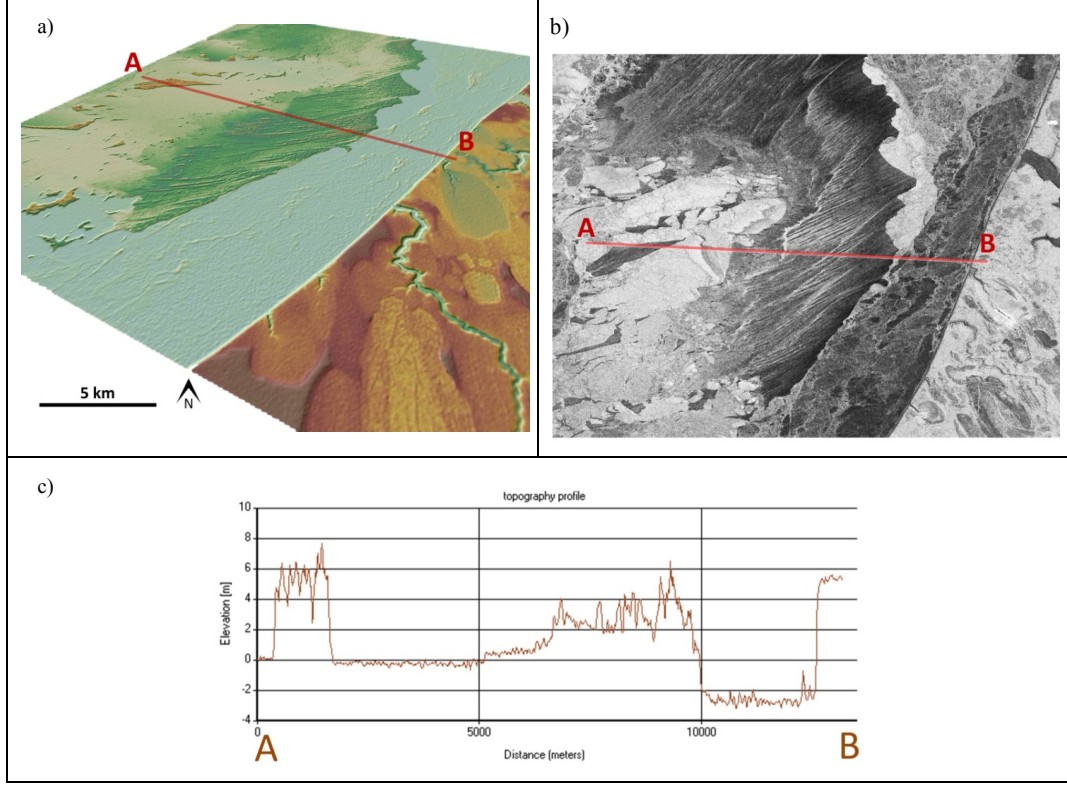

**Figure 4**: (a) Shaded surface topography map from data acquired on March 20, 2015, south-west of Pt. Barrow (subset of
full satellite scene). (b) Corresponding subset of TanDEM-X amplitude image (slant-range geometry). Line AB indicates
5   location of surface topography profile in c) © DLR e.V. 2015 and © Airbus DS Geo GmbH 2015





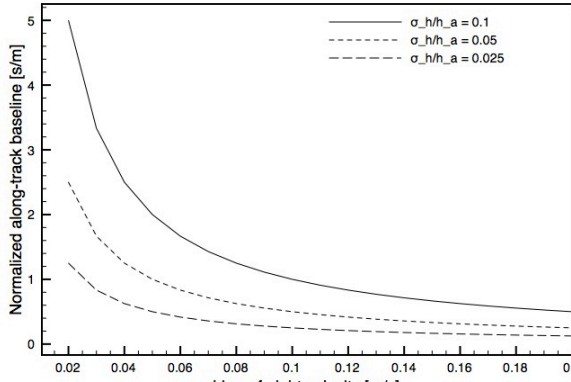

**Figure 5:** "Critical" system-normalized along-track baseline versus the sea ice line-of-sight velocity for different ratios between relative height error and ambiguity height. See text for further explanations.



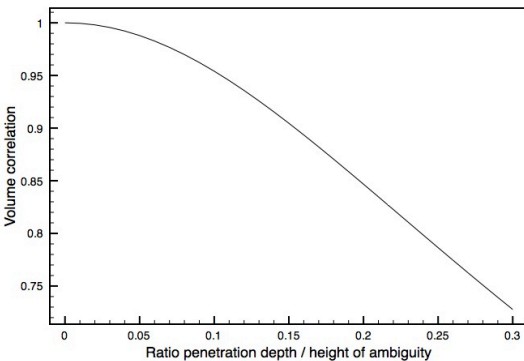

15          **Figure 6.** Volume correlation $|\gamma_{Vol}|$ as a function of the ratio $d/h_{a\_Vol}$ (equation 13).





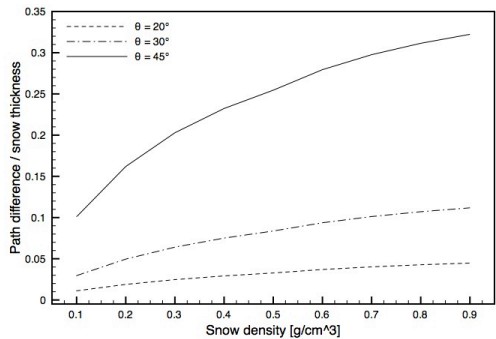

**Figure 7:** Normalized path difference $\Delta s/h$ as a function of snow density $\rho_{ds}$. For explanations, see text.





**Table 1:** Ambiguity heights $h_a$ and relative height errors $\sigma_h$ (rounded values) for optimal baselines $B_n$, determined for different satellite configurations ($\lambda$ – radar wavelength, $H$ – orbit height, $\theta$ – radar incidence angle, $\Delta y$ – ground range resolution). It is assumed that $p=1$, $N_L = 1$, and $\gamma_N \approx 1$ in equations 2, 4, and 8.

| Band | L | | C | | X | | $K_u$ | | $K_a$ | |
|---|---|---|---|---|---|---|---|---|---|---|
| $\lambda$ [m] | 0.24 | | 0.055 | | 0.031 | | 0.022 | | 0.0084 | |
| $H$ [km] | 745 km | | 700 | | 500 | | 780 | | 740 | |
| $\theta$ [deg] | 25 | 40 | 25 | 40 | 25 | 40 | 25 | 40 | 25 | 40 |
| $\Delta y$ [m] | 4.2 | 2.7 | 4.6 | 5.0 | 2.8 | 1.9 | 3.5 | 2.3 | 8.9 | 5.8 |
| $B_{cn}$ [km] | 52 | 112 | 10.2 | 13.1 | 6.7 | 13.9 | 6.0 | 12.7 | 0.85 | 1.8 |
| $B_n$ [km] | 19.8 | 43.1 | 3.9 | 5.0 | 2.6 | 5.3 | 2.3 | 4.9 | 0.32 | 0.69 |
| $h_a$ [m] | 4.2 | 3.5 | 4.6 | 6.4 | 2.8 | 2.4 | 3.5 | 3.0 | 8.9 | 7.5 |
| $\sigma_h$ [m] | 0.60 | 0.50 | 0.66 | 0.92 | 0.40 | 0.35 | 0.50 | 0.42 | 1.3 | 1.1 |



**Table 2:** Noise-equivalent-sigma-zero for different satellite missions

| Satellite Mission | NESZ | Reference |
|---|---|---|
| Tandem-L | -32 dB to -27 dB | Krieger et al., 2010 |
| Sentinel-1 | -22 dB | (https://sentinel.esa.int/web/sentinel/user-guides/sentinel-1-sar) |
| TerraSAR-X / TanDEM-X | -19 dB to -26 dB | (TerraSAR-X Ground Segment Basic Product Specification Document, TX-GS-DD-3302, 2008) |
| Ku-band concept | -24 dB to -29 dB | Lopez-Dekker et al. (2014) |
| SIGNAL | -13 dB | Internal document |





**Table 3:** Ranges of backscattering coefficients $\sigma_0$ for different sea ice types, examples.

| ice type | radar band polarization / incidence angle | $\sigma_0$ – range | Location / sensor / reference |
|---|---|---|---|
| lead ice first-year multi-year | C VV 20-26 deg | -23 to -13 dB -25 to -11 dB -13 to -8 dB | Beaufort Sea ERS-1 Kwok and Cunningham, 1994 |
| young ice first-year brash ridges | C VV (HH) 30-45 deg | -18 to -11 dB *-30 to -15 dB* -14 to -8 dB *-16 to -10 dB* -14 to -6 dB *-15 to -8 dB* -7 to -2 dB *-8 to -4 dB* | Barents Sea, Svalbard Storfjord, Fram Strait airborne SAR (ESAR, DLR) Dierking, 2010 |
| young ice first-year brash ridges | L VV (HH) 30-45 deg | -27 to -14 dB *-29 to -12 dB* -23 to -16 dB *-23 to -16 dB* -14 to -10 dB *-14 to -8 dB* -10 to -6 dB *-10 to -6 dB* | Barents Sea, Svalbard Storfjord, Fram Strait airborne SAR (ESAR, DLR) Dierking, 2010 |
| No distinction | Ku VV+HH merged 20-60 deg | -16 dB to -2 dB | entire Arctic scatterometer Ezraty and Cavanié (1999) |





**Table 4:** Effect of a low signal-to-noise ratio (SNR) on the relative height error $\sigma_h$.

| Band | L | | C | | X | | $K_u$ | | $K_a$ | |
|---|---|---|---|---|---|---|---|---|---|---|
| SNR [dB] | 10 | 5 | 10 | 5 | 10 | 5 | 10 | 5 | 10 | 5 |
| $\sigma_h$ [m] 25° | 0.7 | 0.9 | 0.8 | 1.0 | 0.5 | 0.6 | 0.6 | 0.7 | 1.5 | 1.9 |
| $\sigma_h$ [m] 40° | 0.6 | 0.7 | 1.1 | 1.3 | 0.4 | 0.5 | 0.5 | 0.6 | 1.2 | 1.6 |





**Table 5:** Critical along-track spatial and temporal baselines causing phase shifts corresponding to a height change of 0.5 m, calculated for small and large sea ice drift with the sensor configurations from Table 1. Assumptions: $p = 1$, phase difference per fringe $\Delta\phi_{mov} / 2\pi = 36°$. Because of lacking information we used an orbital velocity over ground of 7 km/s for Tandem-L.

| Band | L | | C | | X | | $K_u$ | | $K_a$ | |
|---|---|---|---|---|---|---|---|---|---|---|
| $v$ [km/s] | 7.0 | | 6.7 | | 7.0 | | 7.0 | | 6.7 | |
| $u_{LOS}$ [m/s] | 0.05 | 0.6 | 0.05 | 0.6 | 0.05 | 0.6 | 0.05 | 0.6 | 0.05 | 0.6 |
| $B_{al}$ [m] | 3360 | 280 | 737 | 61 | 434 | 36 | 308 | 26 | 112 | 9.4 |
| $B_{al}$ [s] | 0.480 | 0.04 | 0.11 | 0.009 | 0.062 | 0.005 | 0.044 | 0.004 | 0.017 | 0.0014 |





**Table 6:** Critical penetration limits $d_c$ for first-year (FY) and multi-year (MY) ice, determined from equation 13 for the satellite constellations shown in Table 1 ($p = 1$). For comparison, typical penetration depths $d$ for FY and MY ice are given

5 (see text). For $d > d_c$, the volume correlation is lower than 0.95. Also shown are heights of ambiguity without ($h_a$) and with ($h_{a\_Vol}$) volume correction according to equation (11), together with the minimum ice thickness $D_{min} = 3.5d$ that is required for equation 13 to be valid. For L-band, equation 14 was applied, with $D = 0.5$ m for FY and $D = 1.5$ m for MY ice.

| Band | L | | C | | X | | $K_u$ | | $K_a$ | |
|---|---|---|---|---|---|---|---|---|---|---|
| $\theta$ [deg] | 25 | 40 | 25 | 40 | 25 | 40 | 25 | 40 | 25 | 40 |
| $h_a$ [m] | 4.2 | 3.4 | 4.6 | 6.4 | 2.8 | 2.4 | 3.5 | 3.0 | 8.9 | 7.5 |
| $h_{a\_Vol}$ [m] MY | 2.7 | 2.4 | 2.9 | 4.6 | 1.8 | 1.7 | 2.2 | 2.2 | 5.7 | 5.4 |
| $h_{a\_Vol}$ [m] FY | 2.4 | 2.2 | 2.6 | 4.2 | 1.6 | 1.6 | 2.0 | 2.0 | 5.1 | 4.9 |
| $d$ [m] MY | *1.0-3.0* | | *0.1-0.6* | | *0.05-0.3* | | *0.03-0.18* | | *0.02-0.08* | |
| $D_{min}$ [m] MY | (3.5-10.5) | | 0.35-2.1 | | 0.18-1.05 | | 0.15-0.63 | | 0.07-0.28 | |
| $d_c$ [m] MY | 0.28 | 0.26 | 0.31 | 0.48 | 0.19 | 0.18 | 0.23 | 0.23 | 0.59 | 0.57 |
| $d$ [m] FY | *0.3-1.0* | | *0.02-0.14* | | *0.01-0.07* | | *0.005-0.05* | | *0.003-0.02* | |
| $D_{min}$ [m] FY | (1.05-3.5) | | 0.07-0.49 | | 0.04-0.25 | | 0.018-0.18 | | 0.01-0.07 | |
| $d_c$ [m] FY | 0.36 | 0.29 | 0.28 | 0.44 | 0.17 | 0.16 | 0.21 | 0.21 | 0.53 | 0.51 |