# Peer review of "Sea ice local surface topography from single-pass satellite InSAR measurements: a feasibility study"

_The Cryosphere, 2017_

## Referee Comment (RC1) · Anonymous Referee #1 · 10 Apr 2017

This paper provides a comprehensive study of the application of spaceborne crosstrack SAR interferometry for the measurement of sea ice topography. The paper anticipates two satellites flying in some form of tandem orbit, and examines both the geometric stereo effect, and the complication of the short time delay between the two image acquisitions. Results are given for potential tandem missions in four frequency bands; L, C, Ku and Ka, reflecting some previous feasibility studies for spaceborne missions at these frequencies. System (noise-equivalent  $\sigma$ 0 (NESZ), incidence angle), orbital (normal and along-track baselines), and environmental ( $\sigma$ 0, penetration and associated volume decorrelation, ice motion, etc.) factors are considered in the analysis. The results are of interest to the sea ice remote sensing community, and the paper

will be a key resource in the evaluation of future tandem InSAR missions which might include sea ice topography as a potential application. I am happy to endorse this paper for inclusion in The Cryosphere but I would like the authors to consider the comments below.

General comments.

1. Would it be possible to measure wave parameters, in particular height, when ocean swell propagates into the pack ice in the marginal ice zone with any of the proposed configurations?

2. Although there is no 'ground truth' for the two examples of ice 'topography' derived from the TanDEM-X, the results in section 4 are still of significant interest and, I think, this section could be improved.

a. the SAR image (Fig. 2b) should be resampled to ground range and the area for which the topography is shown outlined on the image.

b. The increasing azimuth and ground range directions should be marked.

c. The result of interest is the ice topography, so why not show the height directly as a colour coded DEM with a color-bar extending from -1 to 3 m in Fig. 2a? The shaded relief image is nice but not as 'informative' as a more direct illustration of the topography. You are allowed to remove tilts, if necessary!

d. There are areas in the SAR image (Fig. 2b) which suggest variable surface roughness, a profile through the very bright or dark regions would be of interest allowing a comparison between the image radiometry and the large scale roughness.

e. Figure 3a adds little to the science and, as presented, the 'sea ice' radar image in 3b also adds little. However, if the three images (Figs 2a, 2b and 3b) were resampled to ground range with the same scale, the comparison would be interesting. It should be possible to 'see' the same ridges in Fig. 2b and 3b.

TCD
f. Again, I would like the two parts of Fig. 4 to be resampled to ground range so that a direct comparison is possible.

g. As discussed in the text, Fig 4c is very revealing about the problem of line-of-sight motion even when the temporal baseline is 6 milliseconds. Maybe emphasize in the text that this problem is somewhat alleviated at longer wavelengths?

h. In Fig. 4 the ice at 'A' (0-400 m) and  ${\sim}1000\text{-}5000$  m in the profile is very bright in the SAR image but the height variation suggests that the roughness is relatively small scale. Also, there is a marked height change between the shore-fast ice (10000-12500 m) and the ice at 10000-5000 m. Can you comment on this observation?

3. In sections 5.2 to 5.4 the authors, quite legitimately, have concentrated on a quantitative examination of the 'penetration depth', d. In a couple of instances, the height error associated with penetration was estimated as  $\sim$  0.5 d. I think that there should be a clear recognition of the fact that there need not be a simple relation between the penetration depth and the effective horizon in the ice from which the returns appear to be from. For example, at L-band the penetration depth could be significant in cold multiyear ice, but if the ice is relatively uniform in structure then the surface backscattering component could still dominate over the volume component and the effective backscatter horizon could be closer to the surface than 0.5 d. While this is acknowledged in the text I think it could be made clearer.

Some specific comments on the text...

P3L3: 'The length of the across-track baseline determines the sensitivity to height variations...'. Strictly speaking, this should be... 'The component of the across-track baseline perpendicular to the line-of-sight direction determines the sensitivity to height variations...'.

P4L29: 'to be considered: one the one hand'... presumably 'on the one hand'.

P5L10: 'no spectral shift filter is applied.'... Perhaps a suitable reference should be
added here, in case the reader is unaware of this step in some InSAR processing.

P10L10: 'In Figure 5, the "critical system-normalized" along-track baseline Baln = | pBal / v $\lambda$  | is plotted...'. The trouble is that this has units of inverse velocity, not distance. Consequently, I think a better name for this could be 'critical inverse line-of-sight velocity'. Figure 5 would then need to have a different y axis label, although the units are correct, and some rethinking of the following text on page 10 might be necessary.

P10L16-18: Table 3 is referred to twice; this should be Table 5.

P13L29: temperature; missing r.

P14L19: 'elder' is not appropriate, in fact even 'older' is not strictly necessary.

P15L26: 'acceptabe', missing I.

P17L18; the sentence beginning 'This is not made subject of this study, since...', is not clear.

P18L17: 'Doppler shift', insert space.

TCD

---

## Referee Comment (RC2) · A. Mahoney (Referee) · 10 Apr 2017

Summary

This manuscript presents what I believe may be the most comprehensive review of the application of InSAR to the study of sea ice. The authors focus their attention on the use of single-pass InSAR for the measurement of sea ice topography, but their discussion addresses many aspects that are relevant for other sea ice applications of InSAR such as the detection of motion or deformation. These include the physical constraints on useful baselines, incidence angles and radar wavelengths for deriving useful estimates of sea ice surface height. The authors also consider the influence of sea ice type, surface roughness and snow depth on the accuracy of these mea-

surements. In addition to discussing the potential opportunities for InSAR-derived ice topography measurements from existing and future SAR constellations, the manuscript also presents topographic results derived from a bistatic InSAR acquisition over sea ice near Point Barrow, Alaska. Overall, the manuscript is well written and is likely to be an important contribution to the sea ice InSAR literature, particularly as we enter a new era of publicly available data from a growing number of SAR constellations. I have some minor comments about the discussion of elevation results over thin and drifting, which I describe in more detail below, but I believe these should be relatively easy to address.

General comments

1. Clarification of phase interpretation over young and drifting ice In Figure 4a, the authors present surface elevation derived from phase variations over sea ice near Point Barrow. The accompanying SAR amplitude image (Fig 4b) shows a region of landfast ice attached to the coast and separated from drifting sea ice by a lead, which contains bands of frazil. For readers unfamiliar with SAR images of sea ice, it might be helpful to label these features in the amplitude image. In the text, the authors note the "non-negligible height offset" due to surface motion that occurred during the 6 ms temporal baseline between image acquisitions. I recognize that the focus of the paper is on the retrieval of sea ice topography, but this is an interesting and important result that I think would be worth discussing further. For example, in section 5.1 the authors could apply equation 9 to derive the look direction component of ice velocity. This could even be validated using ice velocity measurements from the Barrow sea ice radar (referenced in Fig 3) or from an oceanographic mooring located in the vicinity of point A in Fig 4 (doi:10.18739/A2MT1D).

Also, the authors drawn attention to the apparent roughness of the surface of the lead, which they attribute to "alternating water and frazil stripes", but I feel this explanation could be expanded. The preceding text discusses the phase contribution due to surface currents parallel to the look direction and also makes references to Langmuir

circulation, but it is left to the reader to connect the dots. For readers not familiar with slant-range geometries or Langmuir circulation I would suggest the following changes:

i) label the look direction on Fig 4a and b (see also my note about Fig's 2 and 4 below)

ii) indicate the wind direction, which can be estimated based on the orientation of the frazil bands

iii) state or illustrated how surface currents might vary according to Langmuir circulation.

Specific comments

P1, Line 24:

I would not describe the change in ice surface topography as "steady". Perhaps "near-constant" would be a better phrase.

P2, Line 2:

I feel that "surface roughness" might be amore appropriate term than "surface height variations"

P3, Lines 31-32:

I suggest replacing "neither" with "not" and beginning the sentence with "Also" so that it reads: "Also, if the alongtrack baseline is zero, the interferometric phase is not affected by ice drift".

P8, Line 3:

Note that as of December 1, 2016, the city of Barrow changed its name to UtqiaÄą̇vik. I suggest replacing all references to city of Barrow with its new name and adding "(formerly known as Barrow)" after the first instance. Note that Point Barrow has not changed its name.

P8, Lines 16-17:

Actually, independent measurements of sea ice topography were made for a region of landfast sea ice within the coverage of this DEM. These data are presented in an article by D.O. Dammann recently submitted to the IEEE Journal of Selected Topics in Applied Earth Observations and Remote Sensing.

P11, Line 15:

It should be noted that surface wave amplitudes can be much greater in the marginal ice zone and so this statement should be qualified by noting that infra-gravity waves can be neglected in the central ice pack.

P11, Line 27: According to the stated definition, I believe the authors mean "average height of each sail", rather than "average sail height"

P12, Line 1: I realize that this is partly a matter of style, but I recommend the authors read the following short article on parentheses use:

Robock, A. (2010), Parentheses are (are not) for references and clarification (saving Space), Eos Trans. AGU, 91(45), 419–419, doi:10.1029/2010EO450004.

P12, Lines 1-2: I do not feel this statement is correct in the case where ridges are larger than the resolution cell of the SAR data. While the elevation of the cells near the peak of the sail might be higher than the average height of the overall sail, the cells on the flanks of the sail will have lower-than-average elevations.

P12, Lines 16-18:

Could the authors please provide a citation for these penetration depths?

P13, Line18:

Should this be "2.5" instead of "2-5"?

P14, Line 19:

Correct "elder" to "older"

Figure 2:

It would help the reader interpret these results if the data in panels (a) and (b) could be presented in the same projection.

Figure 4:

As with Figure 2, it would be helpful if the data in panels (a) and (b) could be presented in the same projection. It might also help if there were tick marks at key intervals along the AB transect, with corresponding marks on panel (c).

---

## Author Comment (AC1) · 21 Jun 2017

Please find our answers in the attached files. This editor is not very advanced...

Please also note the supplement to this comment:
http://www.the-cryosphere-discuss.net/tc-2017-40/tc-2017-40-AC1-supplement.pdf

---

## Author Response (AR1)

**Reviewer 1**

Thank you very much for your constructive and encouraging comments.

5   This paper provides a comprehensive study of the application of spaceborne cross-track SAR interferometry for the measurement of sea ice topography. The paper anticipates two satellites flying in some form of tandem orbit, and examines both the geometric stereo effect, and the complication of the short time delay between the two image acquisitions. Results are given for potential tandem missions in four frequency bands; L, C, Ku and Ka,

In fact, we used 5 frequency bands, besides the mentioned ones also X-band

10   reflecting some previous feasibility studies for spaceborne missions at these frequencies. System (noise-equivalent σ0 (NESZ), incidence angle), orbital (normal and along-track baselines), and environmental (σ0, penetration and associated volume decorrelation, ice motion, etc.) factors are considered in the analysis.

The results are of interest to the sea ice remote sensing community, and the paper will be a key resource in the evaluation of future tandem InSAR missions which might include sea ice topography as a potential application. I am happy to endorse this

15   paper for inclusion in The Cryosphere but I would like the authors to consider the comments below.

General comments.

1. Would it be possible to measure wave parameters, in particular height, when ocean swell propagates into the pack ice in

20   the marginal ice zone with any of the proposed configurations?

We think that this questions needs not to be directly addressed in the paper. Since infra-gravity waves with amplitudes of a few millimeters can be recognized in interferograms (see Mahoney et al., Geophys. Res.Lett. 43, 6383-6392, 2016), it may also be possible to measure swell parameters in the marginal ice zone. However, for a comprehensive answer to this question one needs to consider different aspects such as for example: How large is the interferometric decorrelation in the marginal

25   ice zone under different ice and meteorological conditions? Are we talking about a closed sea ice cover? Does the sea ice cover behave like an elastic medium? Is it broken? What time difference between the two images forming the interferometric pair is optimal? Hence we can not provide a simple answer at this point.

2. Although there is no 'ground truth' for the two examples of ice 'topography' derived from the TanDEM-X, the results in

30   section 4 are still of significant interest and, I think, this section could be improved.

a. the SAR image (Fig. 2b) should be resampled to ground range and the area for which the topography is shown outlined on the image.

The SAR image was resampled. The SAR image and the topographic map now show the same area.

b. The increasing azimuth and ground range directions should be marked.

We added this information.

c. The result of interest is the ice topography, so why not show the height directly as a colour coded DEM with a color-bar extending from -1 to 3 m in Fig. 2a? The shaded relief image is nice but not as 'informative' as a more direct illustration of the topography. You are allowed to remove tilts, if necessary!

5   We provided a color-coded DEM but with a different scale to preserve also the information on height variations on land (after some experiments with different color scales).

d. There are areas in the SAR image (Fig. 2b) which suggest variable surface roughness, a profile through the very bright or dark regions would be of interest allowing a comparison between the image radiometry and the large scale roughness.

The radar brightness is not directly related to the (meter-scale) surface topography. We addressed this item in the third

10   paragraph of section 4.

e. Figure 3a adds little to the science and, as presented, the 'sea ice' radar image in 3b also adds little. However, if the three images (Figs 2a, 2b and 3b) were resampled to ground range with the same scale, the comparison would be interesting. It should be possible to 'see' the same ridges in Fig. 2b and 3b.

Unfortunately it is too time-consuming to find a corresponding match between Figs. 2 and an image of the coastal radar and

15   present them at the same projection. Nevertheless we would like to keep Figure 3 since it is instructive for readers not familiar with sea ice conditions.

f. Again, I would like the two parts of Fig. 4 to be resampled to ground range so that a direct comparison is possible.

Please see answer to comment 2a.

g. As discussed in the text, Fig 4c is very revealing about the problem of line-of-sight motion even when the temporal

20   baseline is 6 milliseconds. Maybe emphasize in the text that this problem is somewhat alleviated at longer wavelengths?

We are not sure in what sense the "problem" is alleviated ? In fact, at L-band, the decorrelation time is larger by a factor of 10 compared to X-band. This means that at L-band the phase differences caused by surface water currents can be measured even at larger temporal separations between the two images forming the interferometric pairs than at X-band. We mention this point in the discussion of the second example, last paragraph of section 2.

25   h. In Fig. 4 the ice at 'A' (0-400 m) and   1000-5000 m in the profile is very bright in the SAR image but the height variation suggests that the roughness is relatively small scale.

Please see answer to comment 2d.

Also, there is a marked height change between the shore-fast ice (10000-12500m) and the ice at 10000-5000 m. Can you comment on this observation?

3. In sections 5.2 to 5.4 the authors, quite legitimately, have concentrated on a quantitative examination of the 'penetration depth', d. In a couple of instances, the height error associated with penetration was estimated as

0.5 d. I think that there should be a clear recognition of the fact that there need not be a simple relation between the penetration depth and the effective horizon in the ice from which the returns appear to be from. For example, at L-band the penetration depth could be significant in cold multiyear ice, but if the ice is relatively uniform in structure then the surface backscattering component could still dominate over the volume component and the effective backscatter horizon could be

5    closer to the surface than 0.5 d. While this is acknowledged in the text I think it could be made clearer.

We rewrote the last paragraph of section 5.3 to emphasize the uncertainties one has to face when estimating the elevation bias and retrieval error caused by varying penetration depths and volume decorrelations.

10    Some specific comments on the text

We considered all comments below – the corresponding changes are marked in the manuscript.

P3L3: 'The length of the across-track baseline determines the sensitivity to height variations…' Strictly speaking, this should be 'The component of the across-track baseline perpendicular to the line-of-sight direction determines the sensitivity to

15    height variations ...'.

P4L29: 'to be considered: one the one hand'… presumably 'on the one hand'.

P5L10: 'no spectral shift filter is applied.'...

20    Perhaps a suitable reference should be added here, in case the reader is unaware of this step in some InSAR processing.

Reference was added

P10L10: 'In Figure 5, the "critical system-normalized" along-track baseline Baln = |pBal / vλ| is plotted...'.

The trouble is that this has units of inverse velocity, not distance. Consequently, I think a better name for this could be

25    'critical inverse line-of-sight velocity'. Figure 5 would then need to have a different y axis label, although

the units are correct, and some rethinking of the following text on page 10 might be necessary.

Very good point! Thank you very much for calling our attention to this inconsistency. We separated baseline and the "system coefficient" p/vλ and modified the discussion accordingly, see third paragraph in section 5.1.

30    P10L16-18: Table 3 is referred to twice; this should be Table 5.

P13L29: temperature; missing r.

P14L19: 'elder' is not appropriate, in fact even 'older' is not strictly necessary.

P15L26: 'acceptabe', missing l.

P17L18; the sentence beginning 'This is not made subject of this study, since...', is not clear.

5    We rephrased this sentence.

P18L17: 'Doppler shift', insert space.

**Reviewer 2**

Thank you very much for your constructive and encouraging comments.

15   Summary

This manuscript presents what I believe may be the most comprehensive review of the application of InSAR to the study of sea ice. The authors focus their attention on the use of single-pass InSAR for the measurement of sea ice topography, but their discussion addresses many aspects that are relevant for other sea ice applications of InSAR such as the detection of motion or deformation. These include the physical constraints on useful baselines, incidence angles and radar wavelengths

20   for deriving useful estimates of sea ice surface height. The authors also consider the influence of sea ice type, surface roughness and snow depth on the accuracy of these measurements. In addition to discussing the potential opportunities for InSAR-derived ice topography measurements from existing and future SAR constellations, the manuscript also presents topographic results derived from a bistatic InSAR acquisition over sea ice near Point Barrow, Alaska. Overall, the manuscript is well written and is likely to be an important contribution to the sea ice InSAR literature, particularly as we

25   enter a new era of publicly available data from a growing number of SAR constellations. I have some minor comments about the discussion of elevation results over thin and drifting, which I describe in more detail below, but I believe these should be relatively easy to address.

General comments

30   1. Clarification of phase interpretation over young and drifting ice

In Figure 4a, the authors present surface elevation derived from phase variations over sea ice near Point Barrow. The accompanying SAR amplitude image (Fig 4b) shows a region of landfast ice attached to the coast and separated from drifting sea ice by a lead, which contains bands of frazil. For readers unfamiliar with SAR images of sea ice, it might be helpful to label these features in the amplitude image. In the text, the authors note the "non-negligible height offset" due to

surface motion that occurred during the 6 ms temporal baseline between image acquisitions. I recognize that the focus of the paper is on the retrieval of sea ice topography, but this is an interesting and important result that I think would be worth discussing further. For example, in section 5.1 the authors could apply equation 9 to derive the look direction component of ice velocity. This could even be validated using ice velocity measurements from the Barrow sea ice radar (referenced in Fig 3) or from an oceanographic mooring located in the vicinity of point A in Fig 4(doi:10.18739/A2MT1D).

We agree that the effect of open water surface currents and of sea ice drift on the interferometric phase is an interesting topic that deserves more detailed research. However, in the context of this paper, and considering the lack of complementary information we decided not to include a discussion concerning the derivation of the line-of-sight velocity field over the open water areas and the drift speed of the ice.  The spatial variation of open water surface currents influenced by Langmuir circulation is rather complex. We provide a short description of Langmuir circulation in the second example, last paragraph of section 4. We also calculated ice drift assuming that the elevation difference between the landfast ice (10000-14000) and the drifting ice (1000-5000) is caused by the movement of the latter. However, the resulting velocities are rather large and seem unrealistic. The text for example 2 in section 4 was completely modified.

Also, the authors drawn attention to the apparent roughness of the surface of the lead, which they attribute to "alternating water and frazil stripes", but I feel this explanation could be expanded. The preceding text discusses the phase contribution due to surface currents parallel to the look direction and also makes references to Langmuir circulation, but it is left to the reader to connect the dots. For readers not familiar with slant-range geometries or Langmuir circulation I would suggest the following changes:

i) label the look direction on Fig 4a and b (see also my note about Fig's 2 and 4 below)

On the new versions of Fig. 2 and Fig. 4, arrows indicate azimuth and range direction.

ii) indicate the wind direction, which can be estimated based on the orientation of the frazil bands

We mention in the new text that the streaks are parallel to the wind direction.

iii) state or illustrated how surface currents might vary according to Langmuir circulation.

The text of the second example was extended to provide some information about surface currents formed and influenced by Langmuir circulation (see above).

Specific comments

P1, Line 24:

I would not describe the change in ice surface topography as "steady". Perhaps "near-constant" would be a better phrase.

We used the notation "intermittent".

P2, Line 2:

I feel that "surface roughness" might be amore appropriate term than "surface height variations"

We used "surface height variations" on purpose to make clear that here we focus on elevation changes on the order of decimeters to meters. "Surface roughness" includes the small-scale roughness in the mm- to cm-range. This is now explained explicitly in the text.

P3, Lines 31-32:

I suggest replacing "neither" with "not" and beginning the sentence with "Also" so that it
reads: "Also, if the alongtrack baseline is zero, the interferometric phase is not affected
by ice drift".

We rephrased the sentence completely.

P8, Line 3:

Note that as of December 1, 2016, the city of Barrow changed its name to UtqiaĂ¸avik.
I suggest replacing all references to city of Barrow with its new name and adding
"(formerly known as Barrow)" after the first instance. Note that Point Barrow has not
changed its name.

In the text we give a hint to the new name.

P8, Lines 16-17:

Actually, independent measurements of sea ice topography were made for a region of landfast sea ice within the coverage of this DEM. These data are presented in an article by D.O. Dammann recently submitted to the IEEE Journal of Selected Topics in Applied Earth Observations and Remote Sensing.

Thank you for this information. As additional information: At the University of Tromsø one of the authors (Dierking) is involved in a study in which retrievals of sea ice elevation from interferometric data are compared to data from a scanning laser and stereo photography.

P11, Line 15:

It should be noted that surface wave amplitudes can be much greater in the marginal ice zone and so this statement should be qualified by noting that infra-gravity waves can be neglected in the central ice pack.

We noted it.

P11, Line 27: According to the stated definition, I believe the authors mean "average height of each sail", rather than "average sail height"

We took the notation "average sail height" from the paper by Strub-Klein and Sudom, but we now specified it as suggested.

P12, Line 1: I realize that this is partly a matter of style, but I recommend the authors read the following short article on parentheses use:

Robock, A. (2010), Parentheses are (are not) for references and clarification (saving Space), Eos Trans. AGU, 91(45), 419–419, doi:10.1029/2010EO450004.

Thanks for this hint. We considered it in some but not in all cases. Sometimes the use of parentheses is simply practical…

P12, Lines 1-2: I do not feel this statement is correct in the case where ridges are larger than the resolution cell of the SAR data. While the elevation of the cells near the peak of the sail might be higher than the average height of the overall sail, the cells on the flanks of the sail will have lower-than-average elevations.

Very good point! We modified the text accordingly.

5   P12, Lines 16-18:

Could the authors please provide a citation for these penetration depths?

All are from Lewis et al. (1987), which is now clarified in the text.

P13, Line18:

Should this be "2.5" instead of "2-5"?

10   Corrected.

P14, Line 19:

Correct "elder" to "older"

Following the suggestion of reviewer 1, we don't use "elder" or "older".

Figure 2:

15   It would help the reader interpret these results if the data in panels (a) and (b) could be

presented in the same projection.

In the new Figs. 2 and 4 the projections of the topographic map and the SAR image are the same.

Figure 4:

As with Figure 2, it would be helpful if the data in panels (a) and (b) could be presented in the same projection. It might also

20   help if there were tick marks at key intervals along the AB transect, with corresponding marks on panel (c).

Tick marks used in (c) are shown on the profile lines in (a).

[revised manuscript text omitted]

Wolfgang Dierking 20.6.2017 09:03
**Kommentar [7]:** Reviewer 2

Wolfgang Dierking 20.6.2017 13:51
**Kommentar [8]:** Profiles were placed at different locations.

Wolfgang Dierking 20.6.2017 13:49

Wolfgang Dierking 20.6.2017 09:29
**Kommentar [9]:** Reviewer 1, comment 2h

data was 2.5 m in ground range and 6.6 m in azimuth direction. Assuming that the standard deviation is caused by noise effects and neglecting the correlation between adjacent pixels, the number of looks in the height map is approximately 8.7, and the theoretical relative height error according to equation 8 is between $0.51/\sqrt{8.7}$ to $0.66/\sqrt{8.7}$ m, i. e 0.17 and 0.22 m. The empirical evaluation of a local height statistics hence compares reasonably with the theoretical derivation in section 2.

5     A second example from an area located southwest of Barrow can be found in Figure 4. The data were acquired on March 20, 2015 with a normal baseline of 833 m, an along-track baseline of 42 m, and an incidence angle of 37.2 deg. The height of ambiguity is 14.5 m. The amplitude image (Fig. 4b) reveals that the profile - when starting at point B and moving to the left - crosses a zone of landfast ice (dark grey belt with bright structures), a coastal polynya , i. e. an open water area with indications of wind-driven Langmuir circulation (dark grey area with bright stripes), a narrow zone of thin ice (dark grey zone), and pack ice (bright grey) in which open water leads (dark areas) are embedded. For the retrieved elevation difference between the landfast ice (distance from 11000 to 13000 in Fig. 4c) with elevations between -3 m and -2 m and the drifting pack ice (2500 to 6000) with elevations around zero, we did not find an explanation. As equation (1) reveals, ice movements along the radar line-of-sight cause additional phase differences of the backscattered signal. However, the drift speed of the pack ice calculated from the observed height difference is too large to be realistic. The open water area (distance from 7000 to 11000 m) and the lead (1100 to 2500 m) crossed by the profile AB appear as rugged ice terrain in the height map with heights between two and almost eight meters. We suppose that these apparent height changes are in effect caused by the influence of surface currents in the open water areas. The along-track baseline of 42 m corresponds to a temporal baseline of 6 milliseconds. This time interval is shorter than the decorrelation time of a water surface, which ranges from about 8 to 10 milliseconds at X-band (Romeiser and Thompson, 2000). Hence, the requirement for a measurable phase difference is fulfilled. At L-band, for example, the decorrelation time is larger by a factor of 10 (Romeiser and Thompson, 2000), which means that it is possible to measure phase differences at even larger temporal baselines. The interferometric phase of open water areas is in general proportional to the mean surface current parallel to the radar look direction and contains also contributions associated with the velocity of small wind-induced ripple waves and with the surface currents due to the orbital motion generated by longer ocean waves (Romeiser and Thompson, 2000). In the special situation shown in Fig. 4b, the open water areas bounded by the light blue lines reveal alternating dark and bright strips in the SAR image. This pattern is typical for Langmuir circulation, in which streaks of ice nearly parallel to the wind direction appear on the water surface (Leibovich, 1983). The streaks are visible manifestations of the convergence zones between counterrotating vortices that are present in the near-surface water layer, with their axes of rotation parallel to the wind. The surface current is composed of a component parallel to the streaks and a component perpendicular to them. The former is largest in the zones of convergence and smaller in the zones of divergence. The latter changes direction between neighboring vortices (Leibovich, 1983). The large "height" variations in the open water areas of Fig. 4 may hence be caused by this complex current pattern and possible wave-current interactions. Since the streaks of ice are located in the zones of convergence, their surface is rough (on scales of centimeters), and the backscattered radar intensity is high. Because of lacking complementary data the analysis of Fig. 4 remains on a qualitative level. Nevertheless, the example demonstrates the need to systematically study the influence of open

Wolfgang Dierking 20.6.2017 13:59

Wolfgang Dierking 20.6.2017 15:53

Wolfgang Dierking 21.6.2017 11:43

Wolfgang Dierking 20.6.2017 09:32
Kommentar [10]: Reviewer 1, comment 2h

Wolfgang Dierking 21.6.2017 11:57

Wolfgang Dierking 21.6.2017 11:58

Wolfgang Dierking 21.6.2017 11:58

Wolfgang Dierking 20.6.2017 09:08
Kommentar [11]: Reviewer 1

Wolfgang Dierking 20.6.2017 10:17

Wolfgang Dierking 20.6.2017 11:46
Kommentar [12]: Reviewer 2, comments on Langmuir circulations.

Wolfgang Dierking 20.6.2017 10:50

[revised manuscript text omitted]

Wolfgang Dierking 20.6.2017 08:54
**Kommentar [26]:** Reviewer 1, comments 2a, b,c + reviewer 2

Wolfgang Dierking 19.6.2017 15:09
... [2]
Wolfgang Dierking 19.6.2017 15:29
Wolfgang Dierking 19.6.2017 15:38
Wolfgang Dierking 21.6.2017 11:39
Wolfgang Dierking 20.6.2017 15:50

**Figure 2:** (a) Surface topography map close to the coastline of Pt. Barrow, derived from data acquired on March 29, 2015. Shown is a subset of the full satellite scene. The sampling distance is 12 m. Red polygon: area for estimating the height error (see text). (b) Corresponding TanDEM-X amplitude image in ground-range geometry. The azimuth (Az) and range (Rg) directions of the SAR acquisition are indicated. The line AB in (a) and (b) is the location of the surface topography profile depicted in (c), with corresponding tick marks. © DLR e.V. 2015 and © Airbus DS Geo GmbH 2015

[Figure]

[Figure]

Wolfgang Dierking 20.6.2017 08:55

**Kommentar [27]:** Reviewer 1, comment 2e: We would like to keep this figure to provide an impression for readers not familiar with sea ice conditions

[revised manuscript text omitted]

Wolfgang Dierking 20.6.2017 15:05